# The Implicit Bias of Minima Stability:
# A View from Function Space

**Rotem Mulayoff**
Technion – Israel Institute of Technology
`rotem.mulayof@gmail.com`

**Tomer Michaeli**
Technion – Israel Institute of Technology
`tomer.m@ee.technion.ac.il`

**Daniel Soudry**
Technion – Israel Institute of Technology
`daniel.soudry@gmail.com`

## Abstract

The loss terrains of over-parameterized neural networks have multiple global minima. However, it is well known that stochastic gradient descent (SGD) can stably converge only to minima that are sufficiently flat w.r.t. SGD's step size. In this paper we study the effect that this mechanism has on the function implemented by the trained model. First, we extend the existing knowledge on minima stability to non-differentiable minima, which are common in ReLU nets. We then use our stability results to study a single hidden layer univariate ReLU network. In this setting, we show that SGD is biased towards functions whose second derivative (w.r.t the input) has a bounded weighted $L_1$ norm, and this is regardless of the initialization. In particular, we show that the function implemented by the network upon convergence gets smoother as the learning rate increases. The weight multiplying the second derivative is larger around the center of the support of the training distribution, and smaller towards its boundaries, suggesting that a trained model tends to be smoother at the center of the training distribution.

## 1 Introduction

Understanding the overwhelming success of deep learning requires unveiling the mechanisms that allow over-parametrized models to generalize well, a phenomenon that is in sharp contrast to classical wisdom. It has been suggested that one of the sources for this behavior is the *implicit bias* that training algorithms have towards certain solutions. Implicit biases of generic optimization methods had been known for decades [49] and have also been discussed in the context of neural network training already in [30]. However, their dramatic effect on modern deep learning has only recently started to unveil [36, 57]. For example, in binary classification tasks with linearly separable data, it has been shown that among all global minima, gradient descent (GD) converges to the maximum margin separator [48]. Similarly, in over-parametrized linear regression, GD converges to the minimum norm solution when initialized at zero [57].

Implicit biases were recently studied in many different settings, including in linear convolutional networks [16], matrix and tensor factorization [14, 41], with weight normalization [54] and with various loss functions [15]. Some works have drawn analogies between these biases and traditional regularization schemes. For example, it has been conjectured that the implicit bias of GD can be expressed as some regularization term that is added to the training loss [14]. This turned out to be true for several settings, such as over-parametrized linear regression with the quadratic loss [57], matrix factorization under certain assumptions [26], and linear classification on separable data using losses with an exponential tail [48]. But recent work has pointed out that this is not true in general,

35th Conference on Neural Information Processing Systems (NeurIPS 2021).

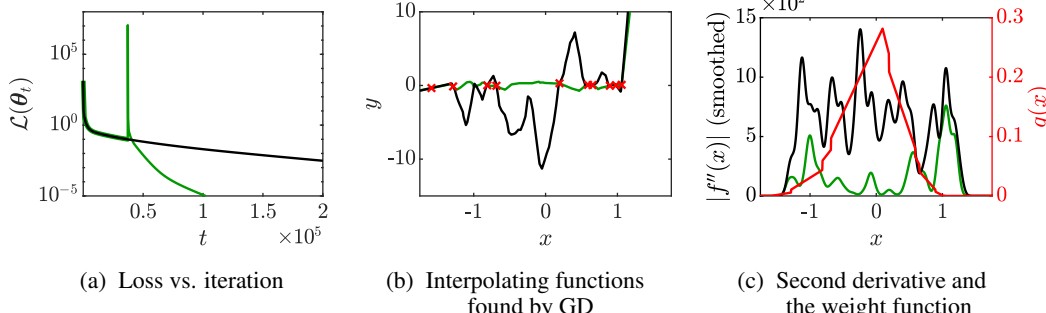

(a) Loss vs. iteration     (b) Interpolating functions found by GD     (c) Second derivative and the weight function

Figure 1: We train a univariate single hidden layer ReLU network using GD. In one experiment (black) we use a constant learning rate $\eta$. In a second experiment (green) we use the same step size and initialization, but when GD arrives near the minimum we increase the step size by a factor of 3. This is equivalent to initializing near the minimum and training with two different step sizes. Panel (a) shows the losses during training. Before the step-size changes, the loss curves coincide. After the change, GD with the larger step-size experiences an abrupt increase in the training loss, and the optimization eventually ends in a different minimum, as evident from Panels (b) and (c). Panel (b) shows the resulting functions implemented by the net, where the cross marks are the training points. The function obtained with the large step size is smoother, in line with our main result (1). Panel (c) depicts (a smoothed version of) $|f''(x)|$ for the two solutions, as well as the weight function $g(x)$ in (1). The $|f''(x)|$ obtained with the larger step-size is smaller, especially where $g(x)$ is high.

by illustrating that the implicit bias of GD is often not equivalent to a loss term of the form of any function of the network's weights [40, 44].

The current paradigm for analyzing implicit bias seeks to ensure convergence to global minima as a first step. This requires making sure that GD is well behaved, *i.e.* does not escape the minimum. A common way to guarantee this is by considering a sufficiently small learning rate, or even the limit of infinitesimal step size, known as gradient flow (GF). In this case, the minima to which GD converges tend to be close to the initialization [39]. Therefore, existing results are initialization-dependent and relevant only to small step-sizes. Unfortunately, these type of analyses fail to capture phenomena that are known to be directly related to the step size in practical training, *e.g.* [8, 17, 19, 24, 29, 45, 46, 53]. For instance, Fig. 1 demonstrates that when initializing GD near a minimum, different step sizes lead to convergence to completely different solutions. This behavior is not reflected by current studies.

Following this understanding, here we seek an initialization-independent analysis that explicitly takes into account the learning rate and reveals its effect on the learned model. To this end, instead of analyzing the trajectory of the parameters from initialization to convergence, we study the properties of the minima to which stochastic GD (SGD) can converge. Specifically, it is well known that SGD cannot stably converge to minima that are too sharp relative to its step size [8, 19, 45, 53]. This property has been studied in [53] for twice differential minima, but its implication on the learned function has not been discussed. Here we extend this analysis to non-differentiable minima, which are common in ReLU networks, and use this property to characterize the end-to-end functions that a neural network can implement upon convergence.

As in [23, 42, 51], we study an over-parameterized single-hidden-layer univariate ReLU network, and focus on the quadratic loss. Under this setting, the network can implement infinitely many piecewise linear functions $f$ that globally minimize the loss (*i.e.* interpolate the data). Our key result (see Theorem 1) is that when using a step size of $\eta$, SGD can only converge to solutions satisfying

$$\int_{\mathbb{R}} |f''(x)|g(x)\mathrm{d}x \leq \frac{1}{\eta} - \frac{1}{2}, \tag{1}$$

with some particular weight function $g$ (see Fig. 1). In other words, the larger the step size, the smoother the solutions that a network learns. The weight $g$ is larger around the center of the support of the training distribution, and smaller towards its boundaries. This implies that the solutions found by SGD tend to be smoother around the center of the distribution.

## 2 Minima stability

SGD is routinely used to minimize objective functions that have multiple global minima. However, it is well known that not all minima are accessible to SGD [53]. Understanding to which minima SGD can converge, requires analyzing its dynamics in the vicinity of minima. If once SGD arrives near a minimum, it converges to it, then we say that this is a stable minimum. If SGD repels from the minimum, then we say that this is an unstable minimum. Here we briefly survey and extend the existing knowledge about stable minima for SGD.

Let $\ell_j : \mathbb{R}^d \mapsto \mathbb{R}$ be differentiable almost everywhere for all $j \in [n]$. Consider a loss function $\mathcal{L}$ and its stochastic counterpart $\hat{\mathcal{L}}$, given by

$$\mathcal{L}(\boldsymbol{\theta}) = \frac{1}{n} \sum_{j=1}^{n} \ell_j(\boldsymbol{\theta}) \qquad \text{and} \qquad \hat{\mathcal{L}}_t(\boldsymbol{\theta}) = \frac{1}{B} \sum_{j \in \mathfrak{B}_t} \ell_j(\boldsymbol{\theta}), \tag{2}$$

where $\mathfrak{B}_t$ is a batch of size $B$ sampled at iteration $t$. We assume that the batches $\{\mathfrak{B}_t\}$ are drawn uniformly from the training set, independently across iterations. SGD's update rule is given by

$$\boldsymbol{\theta}_{t+1} = \boldsymbol{\theta}_t - \eta \nabla \hat{\mathcal{L}}_t(\boldsymbol{\theta}_t), \tag{3}$$

where $\eta$ is the step size. In the following we define the notion of stability for minima, and provide conditions for a minimum to be stable.

### 2.1 Twice differentiable minima

We start by examining a twice differentiable minimum $\boldsymbol{\theta}^*$. Using a Taylor expansion about $\boldsymbol{\theta}^*$, we have

$$\hat{\mathcal{L}}_t(\boldsymbol{\theta}) \approx \hat{\mathcal{L}}_t(\boldsymbol{\theta}^*) + (\boldsymbol{\theta} - \boldsymbol{\theta}^*)^T \nabla \hat{\mathcal{L}}_t(\boldsymbol{\theta}^*) + \frac{1}{2}(\boldsymbol{\theta} - \boldsymbol{\theta}^*)^T \nabla^2 \hat{\mathcal{L}}_t(\boldsymbol{\theta}^*)(\boldsymbol{\theta} - \boldsymbol{\theta}^*), \tag{4}$$

where $\nabla \hat{\mathcal{L}}_t(\boldsymbol{\theta}^*)$ and $\nabla^2 \hat{\mathcal{L}}_t(\boldsymbol{\theta}^*)$ are the gradient and Hessian of $\hat{\mathcal{L}}_t$ at $\boldsymbol{\theta}^*$. Therefore, in the vicinity of the minimum, (3) is approximately given by

$$\boldsymbol{\theta}_{t+1} \approx \boldsymbol{\theta}_t - \eta \left( \nabla \hat{\mathcal{L}}_t(\boldsymbol{\theta}^*) + \nabla^2 \hat{\mathcal{L}}_t(\boldsymbol{\theta}^*)(\boldsymbol{\theta}_t - \boldsymbol{\theta}^*) \right). \tag{5}$$

This approximation gets more accurate as $\boldsymbol{\theta}_t$ gets closer to $\boldsymbol{\theta}^*$. We can thus use this linearized dynamics to learn about the stability of minima [53].

**Definition 1** (Linear stability). *Let $\boldsymbol{\theta}^*$ be a twice differentiable minimum of $\mathcal{L}$. Consider the linearized stochastic dynamical system*

$$\boldsymbol{\theta}_{t+1} = \boldsymbol{\theta}_t - \eta \left( \nabla \hat{\mathcal{L}}_t(\boldsymbol{\theta}^*) + \nabla^2 \hat{\mathcal{L}}_t(\boldsymbol{\theta}^*)(\boldsymbol{\theta}_t - \boldsymbol{\theta}^*) \right). \tag{6}$$

*Then $\boldsymbol{\theta}^*$ is said to be $\varepsilon$ linearly stable if for any $\boldsymbol{\theta}_0$ in the $\varepsilon$-ball $\mathcal{B}_\varepsilon(\boldsymbol{\theta}^*)$, we have $\limsup_{t \to \infty} \mathbb{E}[\|\boldsymbol{\theta}_t - \boldsymbol{\theta}^*\|] \leq \varepsilon$.*

In other words, $\boldsymbol{\theta}^*$ is $\varepsilon$ linearly stable if once we have arrived at a distance of $\varepsilon$ from it (at $t = 0$ without loss of generality), we end up at a distance no greater than $\varepsilon$ around it in expectation. Some of our results do not depend on $\varepsilon$, in which case we simply refer to "linear stability" (without the $\varepsilon$).

Wu et al. [53] provided a sufficient condition for linear stability under the assumption that $\nabla \hat{\mathcal{L}}_t(\boldsymbol{\theta}^*) = \mathbf{0}$ for all $t \geq 1$. For our analysis, we need a necessary condition. This is simple to obtain from the well known stability criterion for GD and the fact that GD's trajectory corresponds to the expectation of SGD's steps. Specifically, we have the following condition (which only requires that $\mathbb{E}[\nabla \hat{\mathcal{L}}_t(\boldsymbol{\theta}^*)] = \mathbf{0}$; see formal proof in Appendix II).

**Lemma 1** (Necessary condition for stability). *Consider SGD with step size $\eta$, where batches are drawn uniformly from the training set, independently across iterations. If $\boldsymbol{\theta}^*$ is an $\varepsilon$ linearly stable minimum of $\mathcal{L}$, then*

$$\lambda_{\max} \left( \nabla^2 \mathcal{L}(\boldsymbol{\theta}^*) \right) \leq \frac{2}{\eta}. \tag{7}$$

Note that this condition is also sufficient when considering GD (*i.e.* full batch SGD).

## 2.2 Non-differentiable minima

We now generalize the definition of linear stability to non-differentiable minima. In deep learning, non-differentiability is typically caused by a mode switch in the system, *e.g.* switching of ReLU activations or max-pooling layers. However, if we fix the mode (*e.g.* constant activation or pooling selection patterns per sample), then the loss becomes infinitely differentiable everywhere. In these cases, we can therefore model SGD's dynamics as a switching dynamical system.

Let $\{\mathcal{S}_m\}$ be a partition of $\mathbb{R}^d$ that represents the regions of the different modes, *i.e.*

$$\forall i \neq j \quad \mathcal{S}_i \cap \mathcal{S}_j = \varnothing, \qquad \text{and} \qquad \bigcup_m \mathcal{S}_m = \mathbb{R}^d. \tag{8}$$

Additionally, let $\boldsymbol{\psi}_m : \mathbb{R}^d \mapsto \mathbb{R}$ be an analytic function representing the loss for the $m$th mode. We assume the overall loss (and its stochastic version) can be written as[1]

$$\mathcal{L}(\boldsymbol{\theta}) = \boldsymbol{\psi}_m(\boldsymbol{\theta}), \qquad \hat{\mathcal{L}}_t(\boldsymbol{\theta}) = \hat{\boldsymbol{\psi}}_m^{(t)}(\boldsymbol{\theta}) \qquad \text{if} \qquad \boldsymbol{\theta} \in \mathcal{S}_m, \tag{9}$$

where $\hat{\boldsymbol{\psi}}_m^{(t)}$ is the stochastic counterpart of $\boldsymbol{\psi}_m$ at time $t$. Therefore, near a minimum $\boldsymbol{\theta}^*$ we can approximate the loss as

$$\hat{\mathcal{L}}_t(\boldsymbol{\theta}) \approx \hat{\mathcal{L}}_t(\boldsymbol{\theta}^*) + (\boldsymbol{\theta} - \boldsymbol{\theta}^*)^T \hat{\boldsymbol{g}}_{\boldsymbol{\theta}}^{(t)} + \frac{1}{2}(\boldsymbol{\theta} - \boldsymbol{\theta}^*)^T \hat{\boldsymbol{H}}_{\boldsymbol{\theta}}^{(t)}(\boldsymbol{\theta} - \boldsymbol{\theta}^*), \tag{10}$$

where (defining $\mathrm{Int}(A)$ as the interior of $A$)

$$\forall \boldsymbol{\theta} \in \mathrm{Int}(\mathcal{S}_m) \qquad \hat{\boldsymbol{g}}_{\boldsymbol{\theta}}^{(t)} \triangleq \nabla \hat{\boldsymbol{\psi}}_m^{(t)}(\boldsymbol{\theta}^*), \qquad \text{and} \qquad \hat{\boldsymbol{H}}_{\boldsymbol{\theta}}^{(t)} \triangleq \nabla^2 \hat{\boldsymbol{\psi}}_m^{(t)}(\boldsymbol{\theta}^*). \tag{11}$$

The update rule of SGD (3) can thus be approximated as

$$\boldsymbol{\theta}_{t+1} \approx \boldsymbol{\theta}_t - \eta \left( \hat{\boldsymbol{g}}_{\boldsymbol{\theta}_t}^{(t)} + \hat{\boldsymbol{H}}_{\boldsymbol{\theta}_t}^{(t)}(\boldsymbol{\theta}_t - \boldsymbol{\theta}^*) \right). \tag{12}$$

Note that this approximation is relevant only when $\boldsymbol{\theta}_t$ is close to $\boldsymbol{\theta}^*$. Let[2] $\mathcal{I} = \{m : \boldsymbol{\theta}^* \in \bar{\mathcal{S}}_m\}$ be the indices of the sets around $\boldsymbol{\theta}^*$, and let $\mathcal{A} = \bigcup_{m \in \mathcal{I}} \mathcal{S}_m$ be their union. We assume there is a finite number of modes in $\mathcal{A}$ (this is always the case for ReLU networks with a finite number of parameters). Consider an $\varepsilon$-neighborhood around the minimum such that $\mathcal{B}_\varepsilon(\boldsymbol{\theta}^*) \subseteq \mathcal{A}$. We define linear stability in this neighborhood as follows.

**Definition 2** (Generalized linear stability)**.** *Let $\boldsymbol{\theta}^*$ be a minimum point of $\mathcal{L}$. Consider the switching stochastic dynamical system*

$$\boldsymbol{\theta}_{t+1} = \boldsymbol{\theta}_t - \eta \left( \hat{\boldsymbol{g}}_{\boldsymbol{\theta}_t}^{(t)} + \hat{\boldsymbol{H}}_{\boldsymbol{\theta}_t}^{(t)}(\boldsymbol{\theta}_t - \boldsymbol{\theta}^*) \right) \tag{13}$$

*and assume that $\boldsymbol{\theta}_t \in \mathcal{A}$ with probability one for all $t > 0$.*

- *$\boldsymbol{\theta}^*$ is said to be $\varepsilon$ linearly stable if $\limsup_{t \to \infty} \mathbb{E}[\|\boldsymbol{\theta}_t - \boldsymbol{\theta}^*\|] \leq \varepsilon$ for any $\boldsymbol{\theta}_0 \in \mathcal{B}_\varepsilon(\boldsymbol{\theta}^*)$.*
- *$\boldsymbol{\theta}^*$ is said to be $\varepsilon$ linearly strongly stable if $\sup_t \mathbb{E}[\|\boldsymbol{\theta}_t - \boldsymbol{\theta}^*\|] \leq \varepsilon$ for any $\boldsymbol{\theta}_0 \in \mathcal{B}_\varepsilon(\boldsymbol{\theta}^*)$.*

In other words, we consider a situation where at some point in time (which we call $t = 0$ without loss of generality), the parameter vector is $\varepsilon$-close to $\boldsymbol{\theta}^*$. If this guarantees that from that moment on, $\boldsymbol{\theta}_t$ is always $\varepsilon$-close to $\boldsymbol{\theta}^*$ in expectation, then $\boldsymbol{\theta}^*$ is called strongly stable. If this only guarantees that from some *later* point in time, $\boldsymbol{\theta}_t$ is always $\varepsilon$-close to $\boldsymbol{\theta}^*$ in expectation, then $\boldsymbol{\theta}^*$ is called stable.

Our results are stated in terms of $\boldsymbol{H}_m = \nabla^2 \boldsymbol{\psi}_m(\boldsymbol{\theta}^*)$ and $\hat{\boldsymbol{g}}_m^{(t)} = \nabla \hat{\boldsymbol{\psi}}_m^{(t)}(\boldsymbol{\theta}^*)$. The next lemma gives a necessary condition for linear stability (see proof in Appendix III).

**Lemma 2** (Necessary condition for stability)**.** *Assume that SGD with step size $\eta$ draws batches uniformly from the training set, independently across iterations. Let $\boldsymbol{\theta}^*$ be a minimum point of $\mathcal{L}$, for which there is a finite number of modes in $\mathcal{I}$. Suppose there exist $\boldsymbol{q} \in \mathbb{S}^{d-1}$ and $\{\lambda_m\}$ such that $\|\boldsymbol{H}_m \boldsymbol{q} - \lambda_m \boldsymbol{q}\| \leq \delta$ for all $m \in \mathcal{I}$ and denote*

$$\lambda^{\mathrm{lower}} = \min_{m \in \mathcal{I}}\{\lambda_m\}, \tag{14}$$

*and[3] $\gamma = \max_{m \in \mathcal{I}} \mathbb{E}[|\boldsymbol{q}^T \hat{\boldsymbol{g}}_m^{(t)}|]$. If*

$$\lambda^{\mathrm{lower}} > \frac{2}{\eta} + \delta + \frac{\gamma}{\varepsilon}, \tag{15}$$

*then $\boldsymbol{\theta}^*$ is not an $\varepsilon$ strongly stable minimum. Furthermore, if $\delta = 0$ (i.e. $\boldsymbol{q}$ is a common eigenvector) then $\boldsymbol{\theta}^*$ is not an $\varepsilon$ stable minimum.*

---

[1]This is true since the modes corresponding to a batch are a subset of those of the full data.

[2]$\bar{\mathcal{S}}_m$ denotes the closure of $\mathcal{S}_m$.

[3]Note that $\{\hat{\boldsymbol{g}}_m^{(t)}\}_{t=1}^\infty$ are i.i.d. and therefore $\mathbb{E}[|\boldsymbol{q}^T \hat{\boldsymbol{g}}_m^{(t)}|]$ is time invariant.

Note that the lemma considers the case where the Hessians of all subsystems have an approximate common eigenvector $\boldsymbol{q}$. In this setting, a necessary condition for stability is that the smallest (approximate) eigenvalue associated with $\boldsymbol{q}$ is not too large w.r.t. $2/\eta$. Interestingly, in the setting of ReLU networks, this (approximate) common eigenvector assumption holds true (see Lemma 5). We thus make use of this property in our analysis of the solutions to which SGD can converge.

For completeness, we also present here a sufficient condition for stability in the special case of GD where $\nabla\boldsymbol{\psi}_m(\boldsymbol{\theta}^*) = \mathbf{0}$ for all $m \in \mathcal{I}$ (see proof in Appendix IV). This setting is of interest in our case, since the global minima of overparameterized ReLU networks under the quadratic loss always satisfy this condition (see proof in Appendix VII for our setting).

**Lemma 3** (Sufficient conditions for stability). *Consider full batch SGD (i.e. GD) with step size $\eta$. Let $\boldsymbol{\theta}^*$ be a minimum point of $\mathcal{L}$, for which $\nabla\boldsymbol{\psi}_m(\boldsymbol{\theta}^*) = \mathbf{0}$ for all $m \in \mathcal{I}$. Denote*

$$\lambda_{\max}^{\text{upper}} = \max_{m \in \mathcal{I}} \lambda_{\max}(\boldsymbol{H}_m). \tag{16}$$

*If*

$$\lambda_{\max}^{\text{upper}} \leq \frac{2}{\eta} \tag{17}$$

*then $\boldsymbol{\theta}^*$ is linearly strongly stable.*

Here we see that if all the subsystems are stable, then the overall switching system is strongly stable.

## 3 The implicit bias of minima stability

We now use our results to study the implicit bias of minima stability in the context of training of univariate ReLU networks. Specifically, consider the set of functions that can be implemented by a one-hidden-layer neural network with $k$ neurons,

$$\mathcal{F} = \left\{ f : \mathbb{R} \mapsto \mathbb{R} \;\middle|\; f(x) = \sum_{i=1}^{k} w_i^{(2)} \sigma\left(w_i^{(1)} x + b_i^{(1)}\right) + b^{(2)} \right\}, \tag{18}$$

where $\sigma(\cdot)$ is the ReLU activation function. Each $f \in \mathcal{F}$ is a piece-wise linear function with at most $k$ knots. We are interested in functions that minimize the quadratic loss,

$$\mathcal{L}(f) = \frac{1}{2n} \sum_{j=1}^{n} \left(f(x_j) - y_j\right)^2, \tag{19}$$

where $\{(x_j, y_j)\}_{j=1}^{n}$ are $n$ training samples such that[4] $x_i \neq x_j$ whenever $i \neq j$.

**Definition 3.** *We say that $f \in \mathcal{F}$ is a solution if $\mathcal{L}(f) = 0$. In this case the function satisfies $f(x_j) = y_j$ for all $j \in [n]$.*

When $k \geq n$ there are infinitely many solutions. However, as discussed above, when training the network using SGD, not all of these global minima can be reached. Particularly, assume we use SGD to minimize the loss w.r.t. the parameter vector

$$\boldsymbol{\theta} = \left[w_1^{(1)}, \ldots, w_k^{(1)}, b_1^{(1)}, \ldots, b_k^{(1)}, w_1^{(2)}, \ldots, w_k^{(2)}, b^{(2)}\right]^T \in \mathbb{R}^{3k+1}. \tag{20}$$

What are the properties of the solutions (in function space) to which we can converge?

In Sec. 2 we saw that dynamic stability is associated with the Hessian of the loss at the minimum. Our goal is thus to link the Hessian to properties of the solution $f$ in function space. A major challenge, however, is that each $f \in \mathcal{F}$ may have infinitely many different implementations $\boldsymbol{\theta}$, and each such implementation may have a different Hessian. Since we are only interested in the functionality $f$ and not in its particular implementation, we consider a solution to be accessible by SGD if there exists some implementation of that solution which is a stable minimum for SGD (see Definition 1 and 2).

**Definition 4.** *We say that a solution $f \in \mathcal{F}$ is (strongly) stable for step-size $\eta$ if there exists a minimum point $\boldsymbol{\theta}^*$ of the loss that corresponds to $f$, where $\boldsymbol{\theta}^*$ is linearly (strongly) stable for SGD with step-size $\eta$.*

---

[4]In this work we focus on cases in which perfect fit is possible. Note that when $x_i = x_j$, if $y_i \neq y_j$ then perfect interpolation is impossible, and if $y_i = y_j$ then the $j$th training sample is redundant.

Our first result characterizes solutions in $\mathcal{F}$ that correspond to twice differentiable minima. For this type of minima, the knots of $f$ do not coincide with any training point in the data set. In the following, $f''(x)$ should be interpreted in the weak sense (*i.e.* it is a sum of weighted Dirac delta functions).

**Theorem 1** (Properties of twice differentiable stable solutions). *Let $f$ be a linearly stable solution for SGD with step-size $\eta$. Assume that the knots of $f$ do not coincide with any training point. Then*

$$\int_{-\infty}^{\infty} |f''(x)| \, g(x) \mathrm{d}x \le \frac{1}{\eta} - \frac{1}{2}, \tag{21}$$

*where*

$$g(x) = \begin{cases} \min\left\{g^-(x), g^+(x)\right\}, & x \in [x_{\min}, x_{\max}], \\ 0, & otherwise, \end{cases} \tag{22}$$

*with*

$$g^-(x) = \mathbb{P}^2 \left(X < x\right) \mathbb{E}\left[x - X | X < x\right] \sqrt{1 + \left(\mathbb{E}[X | X < x]\right)^2},$$

$$g^+(x) = \mathbb{P}^2 \left(X > x\right) \mathbb{E}\left[X - x | X > x\right] \sqrt{1 + \left(\mathbb{E}[X | X > x]\right)^2}. \tag{23}$$

*Here $X$ is drawn from the empirical distribution of the data (a sample chosen uniformly from $\{x_j\}$).*

This theorem shows that stable solutions of SGD correspond to functions whose second derivative has a bounded weighted norm. Importantly, the bound depends on the step size. As the learning rate increases, the set of stable solutions contains less and less non-smooth functions. Figure 2 depicts the weight $g$ for various distributions of the training data. We can see that most of $g$'s mass is located at the center of the training data. Furthermore, $g$ decays towards the extreme data points, and vanishes beyond them. This implies that stable solutions tend to be smoother for instances near the center of the data distribution, and less smooth for instances near the edges. Particularly, minima stability imposes no restrictions on the function's smoothness outside the support of the data distribution.

A limitation of Theorem 1 is that when training with a very large step size, the set of stable solutions contains only very smooth interpolating functions. But such functions tend to have their knots coincide with data points [42], which contradicts the theorem's assumption that the minimum is twice differentiable. To cope with such settings, we now present a result for non-differentiable minima. This result requires assumptions on the maximal number of neurons that can coincide with each data point, and is thus stated in terms of the number of knots of the function $f$ (see Sec. 4.2 for detail).

**Theorem 2** (Properties of non-differentiable strongly stable solutions). *Let $f \in \mathcal{F}$ be a piece-wise linear function with $L \le k$ knots. Assume that $f$ has an implementation $\boldsymbol{\theta}^*$ such that $\|\boldsymbol{\theta}^*\|_\infty \le \rho$. Define $M = \max_{j \in [n]} |x_j|$ and $C = \max\{1, \rho(1 + M)\}$. If $f$ is a linearly strongly stable solution for SGD with step-size $\eta$, then*

$$\int_{-\infty}^{\infty} |f''(x)| \, g(x) \mathrm{d}x \le \frac{1}{\eta} - \frac{1}{2} + \Delta, \tag{24}$$

*where $g(x)$ is defined in (22) and*

$$\Delta = \frac{3}{2} C^2 \frac{1 + k - L}{n} + C \sqrt{\frac{3}{4} \frac{1 + k - L}{n} \left(3 C^2 \frac{1 + k - L}{n} + \frac{2}{\eta}\right)}. \tag{25}$$

Note that in the small over-parametrization regime, $\Delta$ is small. Indeed, $k - L$ is the number of excess neurons, *i.e.* the number of neurons employed by the network beyond the minimum required to realize an $L$-knot function. Therefore, in the regime where the number of samples $n$ and number of neurons $k$ grow while $(k - L)/n \to 0$, we have that $\Delta \to 0$ provided that $C$ is bounded. The latter happens, for example, when the training data are bounded and generated by a smooth function[5].

## 4 Proof outlines

### 4.1 Twice differentiable minima

Our goal is to characterize the minima to which SGD with step size $\eta$ can converge, by using Lemma 1. We start by computing the Hessian matrix at a twice differentiable global minimum with zero error

---

[5]Savarese et al. [42] showed that $\|\boldsymbol{\theta}^*\|_2$ is bounded in this case, implying that $\|\boldsymbol{\theta}^*\|_\infty \le \rho$ as well.

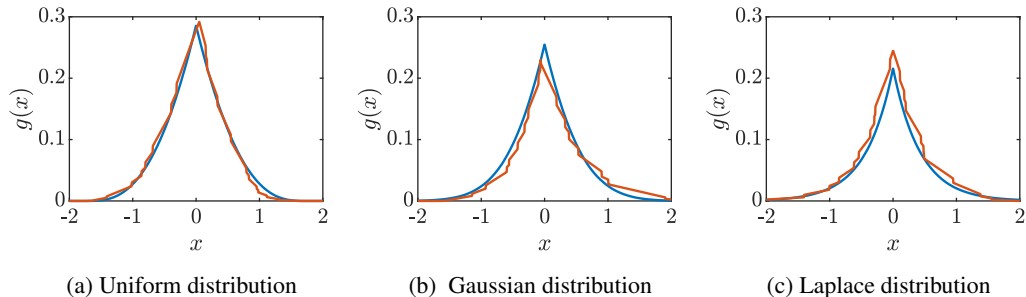

(a) Uniform distribution       (b) Gaussian distribution       (c) Laplace distribution

Figure 2: Graphs of $g(x)$ for different distributions. For each distribution, the empirical graph of $g$ (red) is based on $n = 20$ i.i.d. samples, where we normalized them to zero average and standard deviation one. The theoretical graph (blue) is computed by using the population distribution of $X$ in $g$ (via numerical integration).

(so $f(x_j) = y_j$ for all $j \in [n]$). The gradient of the loss (19) w.r.t. $\boldsymbol{\theta}$ is given by

$$\nabla_{\boldsymbol{\theta}}\mathcal{L} = \frac{1}{n}\sum_{j=1}^{n}\left(f(x_j) - y_j\right)\nabla_{\boldsymbol{\theta}}f(x_j), \tag{26}$$

where $\nabla_{\boldsymbol{\theta}}f(x)$ is written explicitly in Appendix V. Thus, the Hessian is given by

$$\nabla_{\boldsymbol{\theta}}^2\mathcal{L} = \frac{1}{n}\sum_{j=1}^{n}\left(\nabla_{\boldsymbol{\theta}}f(x_j)\right)\left(\nabla_{\boldsymbol{\theta}}f(x_j)\right)^T + \frac{1}{n}\sum_{j=1}^{n}\left(f(x_j) - y_j\right)\nabla_{\boldsymbol{\theta}}^2 f(x_j)$$

$$= \frac{1}{n}\sum_{j=1}^{n}\left(\nabla_{\boldsymbol{\theta}}f(x_j)\right)\left(\nabla_{\boldsymbol{\theta}}f(x_j)\right)^T, \tag{27}$$

where we used the fact that $f(x_j) = y_j$ for all $j \in [n]$. Let us denote the tangent features matrix by $\boldsymbol{\Phi} = [\nabla_{\boldsymbol{\theta}}f(x_1), \nabla_{\boldsymbol{\theta}}f(x_2), \ldots, \nabla_{\boldsymbol{\theta}}f(x_n)]$. Then the Hessian can be expressed as $\nabla_{\boldsymbol{\theta}}^2\mathcal{L} = \boldsymbol{\Phi}\boldsymbol{\Phi}^T/n$, and its maximal eigenvalue can be written as

$$\lambda_{\max}(\nabla_{\boldsymbol{\theta}}^2\mathcal{L}) = \max_{\boldsymbol{v}\in\mathbb{S}^{3k}}\boldsymbol{v}^T\left(\nabla_{\boldsymbol{\theta}}^2\mathcal{L}\right)\boldsymbol{v} = \max_{\boldsymbol{v}\in\mathbb{S}^{3k}}\frac{1}{n}\|\boldsymbol{\Phi}^T\boldsymbol{v}\|^2 = \max_{\boldsymbol{u}\in\mathbb{S}^{n-1}}\frac{1}{n}\|\boldsymbol{\Phi}\boldsymbol{u}\|^2. \tag{28}$$

Notice that this eigenvalue is implementation dependent. Namely, there are uncountably many sets of network parameters which correspond to the same end-to-end function $f \in \mathcal{F}$, and different parameter sets can have different top Hessian eigenvalues. However, recall from Definition 4 that we only care whether there exists one implementation of $f \in \mathcal{F}$ whose top eigenvalue is small enough to allow convergence of SGD to that minimum. We would therefore like to analyze the implementation that minimizes $\lambda_{\max}$. Mathematically, we denote the set of all parameters corresponding to $f$ as

$$\Omega(f) = \left\{\boldsymbol{\theta}\in\mathbb{R}^{3k+1}\,\middle|\,f(x) = \sum_{i=1}^{k}w_i^{(2)}\sigma\left(w_i^{(1)}x + b_i^{(1)}\right) + b^{(2)}\right\}. \tag{29}$$

By using the right-hand side of (28) for $\lambda_{\max}$, we show the following (see proof in Appendix VI).

**Lemma 4** (Top eigenvalue lower bound). *Let $f \in \mathcal{F}$ be a twice-differentiable minimizer of the loss function, then*

$$\min_{\boldsymbol{\theta}\in\Omega(f)}\lambda_{\max}(\nabla_{\boldsymbol{\theta}}^2\mathcal{L}) \geq 1 + 2\int_{-\infty}^{\infty}|f''(x)|\,g(x)\mathrm{d}x, \tag{30}$$

*where $g$ is defined in* (22).

Now we can prove Theorem 1 based on Lemma 1 and 4.

*Proof of Theorem 1.* Since $f$ is a stable solution, there exists a linearly stable minimum point $\boldsymbol{\theta}^*$ of $\mathcal{L}$ such that $\boldsymbol{\theta}^* \in \Omega(f)$. Therefore,

$$1 + 2\int_{-\infty}^{\infty}|f''(x)|\,g(x)\mathrm{d}x \leq \min_{\boldsymbol{\theta}\in\Omega(f)}\lambda_{\max}(\nabla_{\boldsymbol{\theta}}^2\mathcal{L}(\boldsymbol{\theta})) \leq \lambda_{\max}(\nabla_{\boldsymbol{\theta}}^2\mathcal{L}(\boldsymbol{\theta}^*)) \leq \frac{2}{\eta}, \tag{31}$$

where the first inequality follows from Lemma 4 and the last inequality follows from Lemma 1. From the leftmost and rightmost sides of (31), we get

$$\int_{-\infty}^{\infty} |f''(x)|\, g(x)\mathrm{d}x \leq \frac{1}{\eta} - \frac{1}{2}, \tag{32}$$

which completes the proof. $\qquad\square$

## 4.2 Non-differentiable minima

A minimum $\boldsymbol{\theta}^*$ that is not twice differentiable corresponds to a network that interpolates the data, while at least one of its knots coincides with a data point. The neuron corresponding to each such knot, divides the parameter space into two regions: one where the neuron is active, and one where it is inactive. At the interior of each such region, the loss function is twice differentiable w.r.t. parameters. If we denote by $p_j$ the number of neurons that toggle precisely on $x_j$, then the total number of toggling neurons for $\boldsymbol{\theta}^*$ is $p = \sum_{j=1}^n p_j$ and the switching system in Definition 2 has $2^p$ relevant[6] "modes". We denote by $\mathcal{H}$ the set of $2^p$ Hessian matrices corresponding to these modes. Additionally, we denote the analytic function representing $f(x_j; \boldsymbol{\theta})$ for the $\boldsymbol{m}$th mode by $\phi(x_j, \boldsymbol{\theta}, \boldsymbol{m}_j)$. For more details, see our switching system formulation in Appendix VII.

Our key observation is that when a neuron switches its mode, it hardly affects the Hessian matrix, let alone its top eigenvector. Therefore, if not too many neurons toggle, then the principal directions of all matrices in $\mathcal{H}$ tend to align (see proof in Appendix VIII).

**Lemma 5.** *Assume that* $\|\nabla_{\boldsymbol{\theta}}\phi(x_j, \boldsymbol{\theta}^*, \boldsymbol{m}_j)\|_\infty \leq C$ *for all* $j \in [n]$ *and* $\boldsymbol{m} \in \mathcal{I}$. *Denote the maximal number of neurons that toggle on any single data point by* $p_{\max} = \max\{p_j\}$. *Let* $\boldsymbol{H}$ *be some matrix in* $\mathcal{H}$ *having a top eigen-pair* $\boldsymbol{q} \in \mathbb{S}^{3k}$ *and* $\lambda_{\max}(\boldsymbol{H})$. *Then for all other* $\tilde{\boldsymbol{H}} \in \mathcal{H}$,

$$\left\|\tilde{\boldsymbol{H}}\boldsymbol{q} - \lambda_{\max}(\boldsymbol{H})\boldsymbol{q}\right\| \leq \sqrt{3}C\left(\sqrt{\lambda_{\max}(\boldsymbol{H})} + \sqrt{\lambda_{\max}(\tilde{\boldsymbol{H}})}\right)\sqrt{\frac{p_{\max}}{n}}. \tag{33}$$

Note that if we choose $\boldsymbol{H}$ to be the one with the largest top eigenvector, $\lambda_{\max}^{\text{upper}}$ (see (16)), then we get from the lemma that for all $\boldsymbol{H}_{\boldsymbol{m}} \in \mathcal{H}$,

$$\|\boldsymbol{H}_{\boldsymbol{m}}\boldsymbol{q} - \lambda_{\max}^{\text{upper}}\boldsymbol{q}\| \leq \sqrt{3}C\left(\sqrt{\lambda_{\max}^{\text{upper}}} + \sqrt{\lambda_{\max}(\boldsymbol{H}_{\boldsymbol{m}})}\right)\sqrt{\frac{p_{\max}}{n}} \leq 2\sqrt{3}C\sqrt{\lambda_{\max}^{\text{upper}}}\sqrt{\frac{p_{\max}}{n}}. \tag{34}$$

To use Lemma 2, we would like to find an explicit expression for this bound. Under the assumption $\|\boldsymbol{\theta}^*\|_\infty \leq \rho$ we have that $\|\nabla_{\boldsymbol{\theta}}\phi(x_j, \boldsymbol{\theta}, \boldsymbol{m}_j)\|_\infty \leq \max\{1, \rho(1+M)\}$, where $M = \max_{j \in [n]} |x_j|$. Thus, $C$ can be taken as $\max\{1, \rho(1+M)\}$. Note that $\hat{\boldsymbol{g}}_{\boldsymbol{m}}^{(t)} = \boldsymbol{0}$ for all $\boldsymbol{m} \in \mathcal{I}$ and $t > 0$ (see Appendix VII). Thus, using Lemma 2 with $\gamma = 0$, $\delta = 2\sqrt{3}C\sqrt{\lambda_{\max}^{\text{upper}}}\sqrt{\frac{p_{\max}}{n}}$ and $\lambda_{\boldsymbol{m}} = \lambda_{\max}^{\text{upper}}$ for all $\boldsymbol{m}$ (such that $\lambda^{\text{lower}} = \lambda_{\max}^{\text{upper}}$), we get that if $\boldsymbol{\theta}^*$ is strongly stable then (see Appendix IX)

$$\lambda_{\max}^{\text{upper}}(\boldsymbol{\theta}^*) \leq \frac{2}{\eta} + 3C^2\frac{p_{\max}}{n} + C\sqrt{3\frac{p_{\max}}{n}\left(3C^2\frac{p_{\max}}{n} + \frac{2}{\eta}\right)}. \tag{35}$$

If $f \in \mathcal{F}$ has $L \leq k$ knots, then $p_{\max} \leq 1 + k - L$. Substituting this upper-bound in (35), the necessary condition becomes

$$\lambda_{\max}^{\text{upper}}(\boldsymbol{\theta}^*) \leq \frac{2}{\eta} + 3C^2\frac{1+k-L}{n} + C\sqrt{3\frac{1+k-L}{n}\left(3C^2\frac{1+k-L}{n} + \frac{2}{\eta}\right)}. \tag{36}$$

Finally, we show in Appendix X that, similarly to the twice differentiable case of Lemma 4, here as well

$$\min_{\boldsymbol{\theta} \in \Omega(f)} \lambda_{\max}^{\text{upper}}(\boldsymbol{\theta}) \geq 1 + 2\int_{-\infty}^{\infty} |f''(x)|\, g(x)\mathrm{d}x. \tag{37}$$

Combining (37) and (36), shows that any strongly stable solution that meets the assumptions must satisfy (24).

---

[6]Modes that are near the minimum, *i.e.* that are in $\mathcal{I}$.

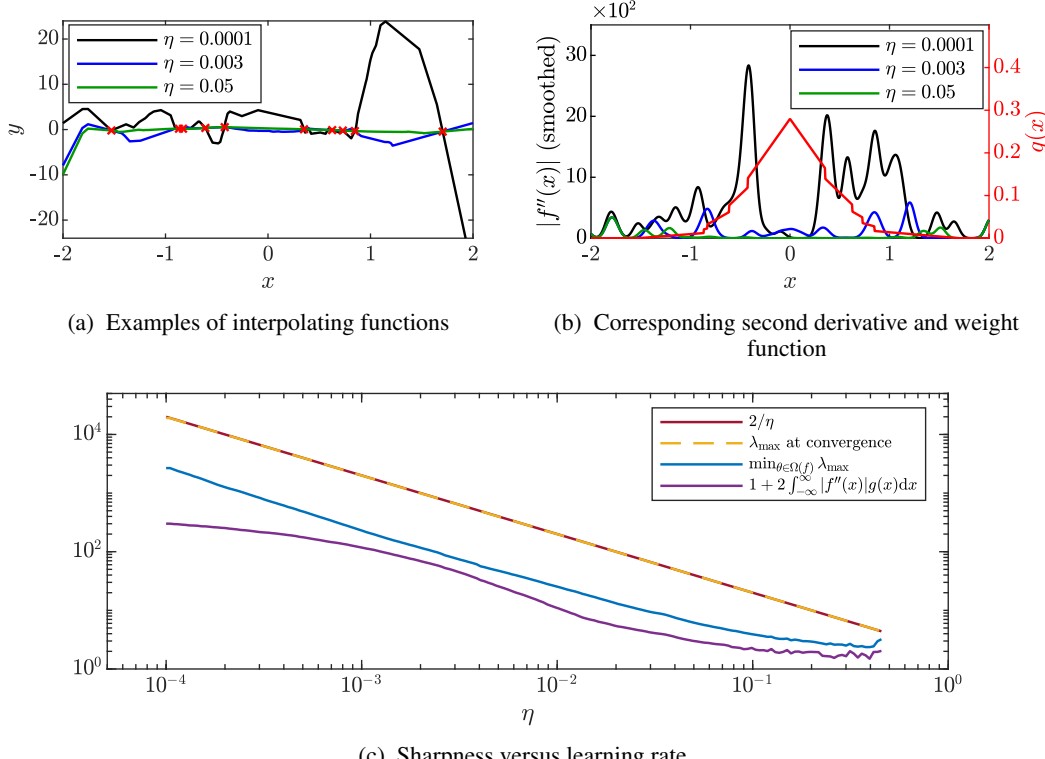

(a) Examples of interpolating functions

(b) Corresponding second derivative and weight function

(c) Sharpness versus learning rate

Figure 3: GD training of a one-hidden-layer ReLU network using various step sizes. Panel (a) shows the training data and the solutions to which GD converged for three different step sizes. Panel (b) depicts the corresponding weight function $g$, and the second derivative of the solutions. Panel (c) visualizes the sharpness of the solutions versus the learning rate. From these plots we see that the GD solution gets smoother as the step size increases, particularly at the center of the training data.

## 5 Experiments

We now verify our theoretical predictions in experiments. We train a single-hidden-layer ReLU network using GD with varying step sizes, all initialized at the same point. Figure 3(a) shows the training data and solutions to which GD converged. Figure 3(b) depicts the corresponding weight function $g$ and the (smoothed) absolute value of the second derivative of the solutions. When trained with a very small step size, GD converges to the black function, whose second derivative is quite large. However, even in this extreme case, the second derivative is small where the weight function $g$ is high, which is in accordance with (21),(24). When trained with a medium step size, GD converges to the blue function, which is much smoother, particularly at the center of the training data. This trend continues as we increase the learning rate. Note that the green function, which is obtained with a large step size, is flat at the center, but has relatively large second derivative outside the support of the training data. This demonstrates that while minima stability restrains the solution within the support of the training data, it imposes no restrictions on the function's smoothness outside of the support.

Figure 3(c) visualizes the sharpness of the solution as a function of the learning rate. Specifically, the red line marks the edge of the stable region, $2/\eta$. Recall that GD cannot converge to a minimum whose sharpness exceeds this value. The yellow dashed line depicts the sharpness of the minima to which GD converged in practice. Here the two lines coincide, indicating that GD converged on the edge of the stable region. This is known to be a common phenomenon [8]. In blue, we plot the sharpness of the flattest implementation for each converged network (see Appendix XI for details). Finally, the purple curve depicts our lower bound on the sharpness of the flattest implementation, presented in Lemma 4. Note that our bound is relatively tight (purple curve close to the blue curve). Moreover, at the large step size regime, it is also quite close to the actual sharpness upon convergence (yellow curve). Additionally, since our bound is the weighted $L_1$ norm of the second derivative of $f$,

the decay of the purple line indicates that $f$ indeed gets smoother as the step size increases, just like our theoretical results predict. Please refer to Appendix I for more details and additional experiments.

## 6 Related work

The implicit regularization of GD was widely studied in the context of matrix factorization and deep linear networks in regression tasks, *e.g.* in [1, 2, 5, 10, 11, 14, 26, 28, 40]. This standard problem was considered to be the key to understanding the inherent bias of GD. Another popular setting which was extensively studied is binary classification of linearly separable data. Soudry et al. [48] showed that among all linear separators that achieve the global minimum of the training loss, GD converges to the maximum margin separator. This result was extended and studied in other settings such as non-separable data, other loss functions, deep linear models, nonlinear networks with homogeneous activations, *etc.* [3, 6, 15, 16, 20–22, 27, 31, 33, 43, 55, 56]. Another long line of works focused on the "Neural Tangent Kernel" regime, which arises with large width or initialization, and a *small* learning rate (e.g., [7, 9, 18, 23, 37]). In this regime, networks converge to a linear predictor minimizing the RKHS norm, where the ("Tangent") kernel is determined by the initialization.

Several works studied theoretically the implicit bias induced by the learning rate. Barrett & Dherin [4] and Smith et al. [47] respectively showed that GD and SGD, for a *small* step-size, approximately follow the GF trajectory on the modified loss, which is the sum of the original loss and a regularization term *on the parameters*. This regularization scales with the step-size and vanishes at any stationary point. Other works focused on linear models, such as linear regression [34, 52], univariate two-layer linear models [25], and deep linear models [32, 35]. Nar & Sastry [35] analyzed deep linear model trained with the quadratic loss and showed that GD cannot converge to certain minima, according to the step size. Mulayoff & Michaeli [32] considered the same setting, and showed that stable solutions (which correspond to flat minima) have special properties. However, for deep linear nets, minima corresponding to different functions, can be equally sharp. This means that minima stability does not promote certain functionalities over others. Accordingly, their results only state that the algorithm has a bias towards certain *implementations*. This is in contrast to our setting, in which we show that SGD is biased towards certain *functions*.

Minima stability was addressed in past work only for twice differentiable minima. Wu et al. [53] proved a sufficient condition under SGD, Tugay & Tanik [50] and Goh [13] gave a stability condition (sufficient and necessary) for GD with Polyak "heavy ball" momentum, Cohen et al. [8] extended his result to Nesterov acceleration, and Giladi et al. [12] derived similar results for asynchronous training. Here we present a new approach for analyzing *non-differentiable minima*, where we provide both necessary and sufficient conditions for stability.

The implicit bias of shallow ReLU networks trained with the quadratic loss, was studied in several works. However, none analyzed the effect of the learning rate. Perhaps most related to our result is [42], which showed that for univariate infinite-width shallow ReLU networks, a solution which minimizes the parameter norm also minimizes $\max(\int |f''(x)|\mathrm{d}x, |f'(\infty) + f'(-\infty)|)$ (this was generalized to multivariate inputs in [38]). However, our result does not rely on the assumption that SGD converges to such a min-norm solution, and it is not yet clear when such an assumption holds. For example, [51] suggests that GF on shallow ReLU nets converges to a min-norm solution only with a vanishing initialization — while for larger initializations, we move closer to a kernel regime, and a very different implicit bias [23, 51]. Additionally, Shamir & Vardi [44] examined GF for a single ReLU neuron, and proved that the implicit bias cannot be *exactly* expressed as a (non-constant) function of the weights. However, bounds like we proved here, do not contradict this result.

## 7 Conclusion

SGD cannot converge stably to any minimum of the loss terrain. Prior work pointed out that stable minima are flat with respect to the step size. Here we presented a new approach for analyzing the stability of non-differentiable minima. Using our results, we examined a simple model of a single hidden layer univariate ReLU network. We showed that in this setting, stable solutions correspond to functions whose second derivative has a bounded weighted $L_1$ norm. Particularly, we showed that the implemented function gets smoother as the learning rate increases, especially near the center of the support of the training distribution. Finally, we demonstrated our theoretical results in experiments.

**Acknowledgements**   The research of Rotem Mulayoff was supported by the Planning and Budgeting Committee of the Israeli Council for Higher Education, and by the Andrew and Erna Finci Viterbi Graduate Fellowship. The research of Tomer Michaeli was supported by the Technion Ollendorff Minerva Center. The research of Daniel Soudry was supported by the Israel Science Foundation (grant No. 1308/18), and by the Israel Innovation Authority (the Avatar Consortium).

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
