# The Implicit Bias of Minima Stability:
# A View from Function Space
# Supplementary Material

**Rotem Mulayoff**
Technion – Israel Institute of Technology
`rotem.mulayof@gmail.com`

**Tomer Michaeli**
Technion – Israel Institute of Technology
`tomer.m@ee.technion.ac.il`

**Daniel Soudry**
Technion – Israel Institute of Technology
`daniel.soudry@gmail.com`

This document contains supplementary material for the article 'The Implicit Bias of Minima Stability: A View from Function Space', and includes the following parts:

## I  Experimental details and additional experiments

For the experiment in Figure 1 we generated $n = 10$ random data points $\{(x_j, y_j)\}_{j=1}^{10}$, and trained a one-hidden-layer ReLU network with $k = 80$ neurons to fit the data. In one experiment (black) we used a constant learning rate of $\eta = 10^{-4}$. In a second experiment (green) we used the same step size and initialization, but when the loss past below $0.1$ for the first time, we increased the step size by a factor of 3. In both experiments we trained the model until $\mathcal{L}(\boldsymbol{\theta}_t) \leq 10^{-8}$.

For the experiment in Sec. 5 we generated a different dataset of $n = 10$ random points, and trained a one-hidden-layer ReLU network with $k = 80$ neurons to fit the data using 200 step sizes in the range $[10^{-4}, 0.5]$. The step sizes were uniformly spaced in a logarithmic scale. We used the same initial point for all training runs, and trained the model until $\mathcal{L}(\boldsymbol{\theta}_t) \leq 10^{-8}$. To be able to converge also with the large step sizes, we used a linear learning rate warm-up for the first $5 \times 10^5$ iterations. This learning rate warm-up was used in all training runs. We excluded solutions for which GD converged during the warm-up period. Therefore, in all runs, the dynamics of GD upon convergence was determined by a constant step size. In particular, out of the 200 step sizes, for 2 step sizes GD converged during the warm-up, and for 15 step sizes the algorithm did not converge to a global

minimum at all. Therefore, overall, Fig. 3 depicts the training of 183 different step sizes. Figures 1 and 2 present additional solutions that GD converged to and their second derivative, respectively. From these figures we see again that GD's solutions get smoother as the step size increases, particularly at the center of the training data.

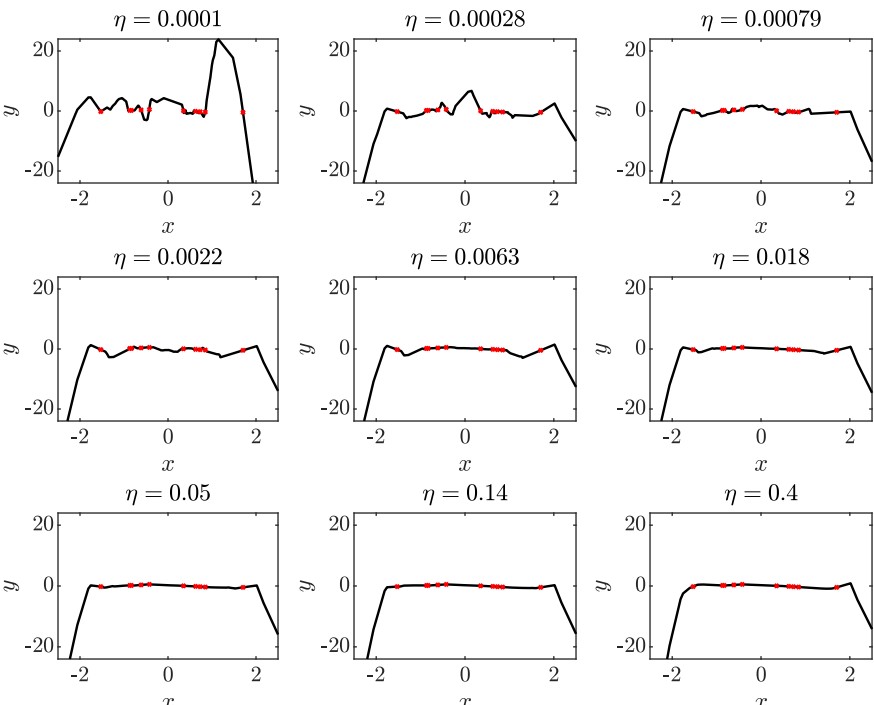

Figure 1: Examples of interpolating functions to which GD converged for various step sizes form the simulation of Sec. 5. Here we see that the GD solutions get smoother as the step size increases, particularly at the center of the training data.

Finally, to examine the effect that initialization magnitude has on the sharpness of the solutions to which GD converges, we repeated the experiment from Sec. 5 using various initialization magnitudes. Specifically, we used PyTorch's standard initialization, and multiplied it by different factors. Then, form each such initial point we trained a single-hidden-layer ReLU network using 100 step sizes in $[10^{-3}, 1.5]$. Figure 3 presents the sharpness curves for the different initialization factors. Here we see that for large initialization, the sharpness monotonically decreases as the learning rate gets larger. Additionally, the function gets smoother as the learning rate increases. However, for moderate initialization this phenomenon occurs only when the learning rate is larger than some threshold. In particular, for all step sizes below this threshold GD converges to a certain minimum that is flat enough w.r.t. these step sizes. Yet, for step sizes above the threshold, this minimum is unstable, and therefore GD converges to flatter minima. Hence, we see that when GD converges at the edge of stability, a small increase in the learning rate translates to a flatter solution. However, if the entire trajectory of GD resides in a smooth region in which the top eigenvalue of the Hessian is smaller than $2/\eta$, then a small increase in the learning rate will not have an effect. Having said that, we note that (a) while our bound may be loose in the latter case, it still holds, and (b) convergence at the edge of stability is rather the typical case [8].

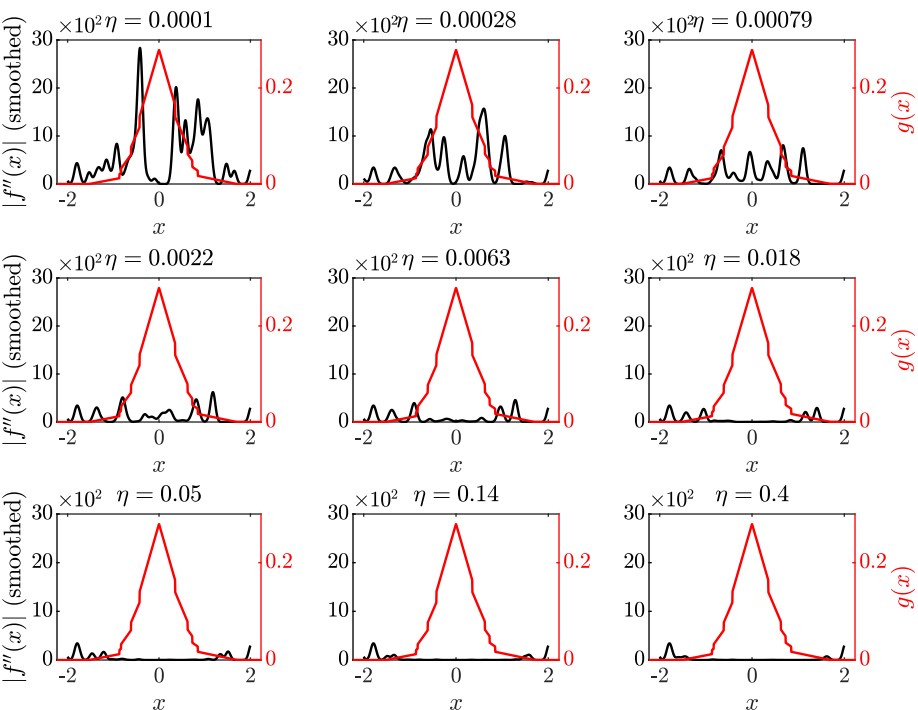

Figure 2: The second derivative of the functions shown in Fig. 1. Here we see that GD solution gets smoother as the step size increases, particularly at the center of the training data where the weight function $g$ is high.

## II Proof of Lemma 1

Under the assumption that SGD draws batches uniformly at random (among the $\binom{n}{k}$ possibilities) from the dataset, we have that for all $\boldsymbol{\theta} \in \mathbb{R}^d$

$$\mathbb{E}\left[\hat{\mathcal{L}}_t(\boldsymbol{\theta})\right] = \mathcal{L}(\boldsymbol{\theta}). \tag{S1}$$

Namely, $\hat{\mathcal{L}}_t(\boldsymbol{\theta})$ is an unbiased estimator of the loss function. Additionally, since we assume that the batches are statistically independent, it follow that $\boldsymbol{\theta}_t$ (which is a function of only $\{\mathfrak{B}_\tau\}_{\tau=1}^{t-1}$) is independent of $\hat{\mathcal{L}}_t(\boldsymbol{\theta})$. Therefore,

$$
\begin{aligned}
\mathbb{E}\left[\boldsymbol{\theta}_{t+1} - \boldsymbol{\theta}^*\right] &= \mathbb{E}\left[\boldsymbol{\theta}_t - \boldsymbol{\theta}^* - \eta\left(\nabla\hat{\mathcal{L}}_t(\boldsymbol{\theta}^*) + \nabla^2\hat{\mathcal{L}}_t(\boldsymbol{\theta}^*)(\boldsymbol{\theta}_t - \boldsymbol{\theta}^*)\right)\right] \\
&= \mathbb{E}\left[\left(\boldsymbol{I} - \eta\nabla^2\hat{\mathcal{L}}_t(\boldsymbol{\theta}^*)\right)(\boldsymbol{\theta}_t - \boldsymbol{\theta}^*)\right] - \eta\mathbb{E}\left[\nabla\hat{\mathcal{L}}_t(\boldsymbol{\theta}^*)\right] \\
&= \mathbb{E}\left[\boldsymbol{I} - \eta\nabla^2\hat{\mathcal{L}}_t(\boldsymbol{\theta}^*)\right]\mathbb{E}\left[\boldsymbol{\theta}_t - \boldsymbol{\theta}^*\right] - \eta\nabla\mathbb{E}\left[\hat{\mathcal{L}}_t(\boldsymbol{\theta}^*)\right] \\
&= \left(\boldsymbol{I} - \eta\nabla^2\mathbb{E}\left[\hat{\mathcal{L}}_t(\boldsymbol{\theta}^*)\right]\right)\mathbb{E}\left[\boldsymbol{\theta}_t - \boldsymbol{\theta}^*\right] - \eta\nabla\mathcal{L}(\boldsymbol{\theta}^*) \\
&= \left(\boldsymbol{I} - \eta\nabla^2\mathcal{L}(\boldsymbol{\theta}^*)\right)\mathbb{E}\left[\boldsymbol{\theta}_t - \boldsymbol{\theta}^*\right],
\end{aligned} \tag{S2}
$$

where in the third step we used the fact that $\boldsymbol{\theta}_t$ and $\hat{\mathcal{L}}_t(\boldsymbol{\theta}^*)$ are independent, whereas in the last step we used the fact that $\nabla\mathcal{L}(\boldsymbol{\theta}^*) = \boldsymbol{0}$. Thus, for all $t \in \mathbb{N}$

$$\mathbb{E}\left[\boldsymbol{\theta}_t - \boldsymbol{\theta}^*\right] = \left(\boldsymbol{I} - \eta\nabla^2\mathcal{L}(\boldsymbol{\theta}^*)\right)^t(\boldsymbol{\theta}_0 - \boldsymbol{\theta}^*). \tag{S3}$$

Observe that according to Jensen's inequality,

$$\left\|\left(\boldsymbol{I} - \eta\nabla^2\mathcal{L}(\boldsymbol{\theta}^*)\right)^t(\boldsymbol{\theta}_0 - \boldsymbol{\theta}^*)\right\| = \|\mathbb{E}\left[\boldsymbol{\theta}_t - \boldsymbol{\theta}^*\right]\| \leq \mathbb{E}\left[\|\boldsymbol{\theta}_t - \boldsymbol{\theta}^*\|\right]. \tag{S4}$$

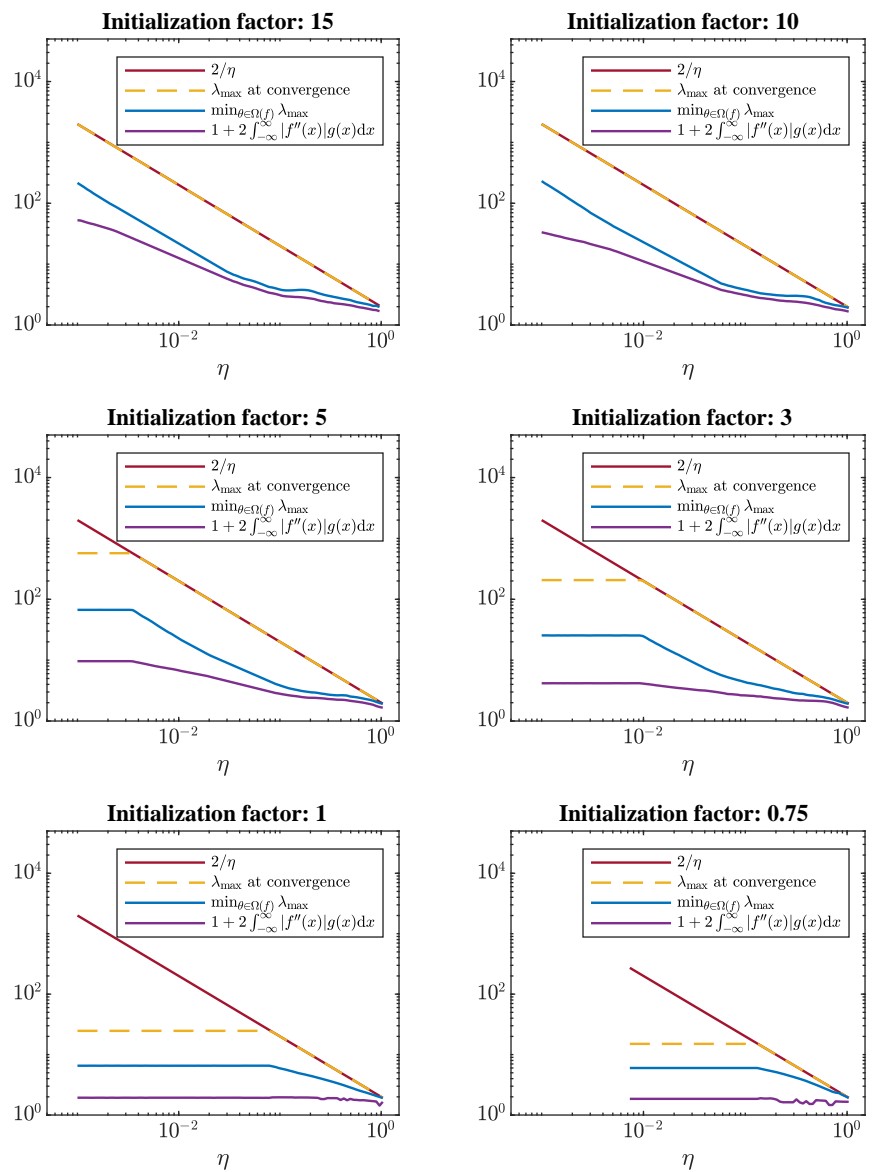

Figure 3: Sharpness curves for various initialization magnitudes

Assuming that $\boldsymbol{\theta}^*$ is a linearly stable point of $\mathcal{L}$, we have that

$$\limsup_{t \to \infty} \mathbb{E}\left[\|\boldsymbol{\theta}_t - \boldsymbol{\theta}^*\|\right] \leq \varepsilon \tag{S5}$$

for all $\boldsymbol{\theta}_0 \in \mathcal{B}_\varepsilon(\boldsymbol{\theta}^*)$. Using (S4) and (S5), we have that

$$\limsup_{t \to \infty} \left\| \left(\boldsymbol{I} - \eta\nabla^2\mathcal{L}(\boldsymbol{\theta}^*)\right)^t (\boldsymbol{\theta}_0 - \boldsymbol{\theta}^*) \right\| \leq \varepsilon. \tag{S6}$$

Let us choose $\boldsymbol{\theta}_0$ such that $(\boldsymbol{\theta}_0 - \boldsymbol{\theta}^*)/\|\boldsymbol{\theta}_0 - \boldsymbol{\theta}^*\|$ is a top eigenvector of $\nabla^2\mathcal{L}(\boldsymbol{\theta}^*)$ and $\|\boldsymbol{\theta}_0 - \boldsymbol{\theta}^*\| = \varepsilon$. Then, we get that

$$\limsup_{t \to \infty} \left| 1 - \eta\lambda_{\max}\left(\nabla^2\mathcal{L}(\boldsymbol{\theta}^*)\right) \right|^t \leq 1. \tag{S7}$$

Therefore,
$$\left|1 - \eta\lambda_{\max}\left(\nabla^2\mathcal{L}(\boldsymbol{\theta}^*)\right)\right| \leq 1, \tag{S8}$$
which implies that
$$\lambda_{\max}\left(\nabla^2\mathcal{L}(\boldsymbol{\theta}^*)\right) \leq \frac{2}{\eta}. \tag{S9}$$

## III  Proof of Lemma 2

With slight abuse of notation, we interchangeably use $\boldsymbol{H}_{\boldsymbol{\theta}}$ and $\boldsymbol{H}_m$, where the meaning in the former is that $\boldsymbol{H}_{\boldsymbol{\theta}} = \boldsymbol{H}_m$ if $\boldsymbol{\theta} \in \mathcal{S}_m$. Similarly, we interchangeably use $\hat{\boldsymbol{g}}_{\boldsymbol{\theta}}$ and $\hat{\boldsymbol{g}}_m$, and $\lambda_{\boldsymbol{\theta}}$ and $\lambda_m$.

Under the assumption that SGD draws batches uniformly at random from the dataset, we have that for all $\boldsymbol{\theta} \in \mathcal{B}_\varepsilon(\boldsymbol{\theta}^*)$
$$\mathbb{E}\left[\hat{\boldsymbol{H}}_{\boldsymbol{\theta}}^{(t)}\right] = \boldsymbol{H}_{\boldsymbol{\theta}}. \tag{S10}$$

Namely, $\hat{\boldsymbol{H}}_{\boldsymbol{\theta}}^{(t)}$ is an unbiased estimator of the Hessian. Additionally, since we assume that the batches are statistically independent, it follows that $\boldsymbol{\theta}_t$ (a function of $\{\hat{\boldsymbol{g}}_{\boldsymbol{\theta}_\tau}^{(\tau)}, \hat{\boldsymbol{H}}_{\boldsymbol{\theta}_\tau}^{(\tau)}\}_{\tau=1}^{t-1}$ solely) is independent of $\{\hat{\boldsymbol{g}}_{\boldsymbol{\theta}}^{(t)}, \hat{\boldsymbol{H}}_{\boldsymbol{\theta}}^{(t)}\}$. Let
$$\gamma = \max_{m\in\mathcal{I}}\mathbb{E}\left[\left|\boldsymbol{q}^T\hat{\boldsymbol{g}}_m^{(t)}\right|\right]. \tag{S11}$$

Assume that $\boldsymbol{\theta}^*$ is a stable minimum (and therefore $\boldsymbol{\theta}_t \in \mathcal{A}$ for all $t > 0$). Let us look at the conditional expectation of $\left|\boldsymbol{q}^T\left(\boldsymbol{\theta}_{t+1} - \boldsymbol{\theta}^*\right)\right|$ given $\boldsymbol{\theta}_t$,

$$\begin{aligned}
\mathbb{E}\left[\left|\boldsymbol{q}^T\left(\boldsymbol{\theta}_{t+1} - \boldsymbol{\theta}^*\right)\right|\Big|\boldsymbol{\theta}_t\right] &= \mathbb{E}\left[\left|\boldsymbol{q}^T\left(\boldsymbol{\theta}_t - \boldsymbol{\theta}^* - \eta\left(\hat{\boldsymbol{g}}_{\boldsymbol{\theta}_t}^{(t)} + \hat{\boldsymbol{H}}_{\boldsymbol{\theta}_t}^{(t)}(\boldsymbol{\theta}_t - \boldsymbol{\theta}^*)\right)\right)\right|\Big|\boldsymbol{\theta}_t\right] \\
&= \mathbb{E}\left[\left|\boldsymbol{q}^T\left(\boldsymbol{I} - \eta\hat{\boldsymbol{H}}_{\boldsymbol{\theta}_t}^{(t)}\right)(\boldsymbol{\theta}_t - \boldsymbol{\theta}^*) - \eta\boldsymbol{q}^T\hat{\boldsymbol{g}}_{\boldsymbol{\theta}_t}^{(t)}\right|\Big|\boldsymbol{\theta}_t\right] \\
&\geq \mathbb{E}\left[\left|\boldsymbol{q}^T\left(\boldsymbol{I} - \eta\hat{\boldsymbol{H}}_{\boldsymbol{\theta}_t}^{(t)}\right)(\boldsymbol{\theta}_t - \boldsymbol{\theta}^*)\right|\Big|\boldsymbol{\theta}_t\right] - \eta\mathbb{E}\left[\left|\boldsymbol{q}^T\hat{\boldsymbol{g}}_{\boldsymbol{\theta}_t}^{(t)}\right|\Big|\boldsymbol{\theta}_t\right] \\
&\geq \left|\mathbb{E}\left[\boldsymbol{q}^T\left(\boldsymbol{I} - \eta\hat{\boldsymbol{H}}_{\boldsymbol{\theta}_t}^{(t)}\right)(\boldsymbol{\theta}_t - \boldsymbol{\theta}^*)\Big|\boldsymbol{\theta}_t\right]\right| - \eta\gamma \\
&= \left|\boldsymbol{q}^T\left(\boldsymbol{I} - \eta\mathbb{E}\left[\hat{\boldsymbol{H}}_{\boldsymbol{\theta}_t}^{(t)}\Big|\boldsymbol{\theta}_t\right]\right)(\boldsymbol{\theta}_t - \boldsymbol{\theta}^*)\right| - \eta\gamma \\
&= \left|\boldsymbol{q}^T\left(\boldsymbol{I} - \eta\boldsymbol{H}_{\boldsymbol{\theta}_t}\right)(\boldsymbol{\theta}_t - \boldsymbol{\theta}^*)\right| - \eta\gamma, \tag{S12}
\end{aligned}$$

where in the first inequality we used the triangle inequality, and in the second we used Jensen's inequality and the definition of $\gamma$ in (S11). Now let $\lambda_{\boldsymbol{\theta}}$ be the (approximate) eigenvalue associated with the Hessian matrix at the region of $\boldsymbol{\theta}$, namely $\lambda_{\boldsymbol{\theta}} = \lambda_m$ for all $\boldsymbol{\theta} \in \mathcal{S}_m$. Under the assumption of the lemma, $\|\lambda_{\boldsymbol{\theta}}\boldsymbol{q} - \boldsymbol{H}_{\boldsymbol{\theta}}\boldsymbol{q}\| \leq \delta$ for all $\boldsymbol{\theta} \in \mathcal{A}$. Therefore

$$\begin{aligned}
\left|\boldsymbol{q}^T\left(\boldsymbol{I} - \eta\boldsymbol{H}_{\boldsymbol{\theta}_t}\right)(\boldsymbol{\theta}_t - \boldsymbol{\theta}^*)\right| &= \left|\boldsymbol{q}^T(\boldsymbol{\theta}_t - \boldsymbol{\theta}^*) - \eta\boldsymbol{q}^T\boldsymbol{H}_{\boldsymbol{\theta}_t}(\boldsymbol{\theta}_t - \boldsymbol{\theta}^*)\right| \\
&= \left|\boldsymbol{q}^T(\boldsymbol{\theta}_t - \boldsymbol{\theta}^*) - \eta\lambda_{\boldsymbol{\theta}_t}\boldsymbol{q}^T(\boldsymbol{\theta}_t - \boldsymbol{\theta}^*) + \eta\lambda_{\boldsymbol{\theta}_t}\boldsymbol{q}^T(\boldsymbol{\theta}_t - \boldsymbol{\theta}^*) - \eta\boldsymbol{q}^T\boldsymbol{H}_{\boldsymbol{\theta}_t}(\boldsymbol{\theta}_t - \boldsymbol{\theta}^*)\right| \\
&= \left|(1 - \eta\lambda_{\boldsymbol{\theta}_t})\boldsymbol{q}^T(\boldsymbol{\theta}_t - \boldsymbol{\theta}^*) + \eta\left(\lambda_{\boldsymbol{\theta}_t}\boldsymbol{q}^T - \boldsymbol{q}^T\boldsymbol{H}_{\boldsymbol{\theta}_t}\right)(\boldsymbol{\theta}_t - \boldsymbol{\theta}^*)\right| \\
&\geq \left|1 - \eta\lambda_{\boldsymbol{\theta}_t}\right|\left|\boldsymbol{q}^T(\boldsymbol{\theta}_t - \boldsymbol{\theta}^*)\right| - \eta\left|\left(\lambda_{\boldsymbol{\theta}_t}\boldsymbol{q}^T - \boldsymbol{q}^T\boldsymbol{H}_{\boldsymbol{\theta}_t}\right)(\boldsymbol{\theta}_t - \boldsymbol{\theta}^*)\right| \\
&\geq \left|1 - \eta\lambda_{\boldsymbol{\theta}_t}\right|\left|\boldsymbol{q}^T(\boldsymbol{\theta}_t - \boldsymbol{\theta}^*)\right| - \eta\left\|\lambda_{\boldsymbol{\theta}_t}\boldsymbol{q} - \boldsymbol{H}_{\boldsymbol{\theta}_t}\boldsymbol{q}\right\|\left\|\boldsymbol{\theta}_t - \boldsymbol{\theta}^*\right\| \\
&\geq \left|1 - \eta\lambda_{\boldsymbol{\theta}_t}\right|\left|\boldsymbol{q}^T(\boldsymbol{\theta}_t - \boldsymbol{\theta}^*)\right| - \eta\delta\left\|\boldsymbol{\theta}_t - \boldsymbol{\theta}^*\right\|, \tag{S13}
\end{aligned}$$

where in the first inequality we used the triangle inequality, in the second we used the Cauchy-Schwarz inequality, and in the last we used the assumption that $\|\lambda_{\boldsymbol{\theta}}\boldsymbol{q} - \boldsymbol{H}_{\boldsymbol{\theta}}\boldsymbol{q}\| \leq \delta$. Hence,

$$\begin{aligned}
\mathbb{E}\left[\left|\boldsymbol{q}^T\left(\boldsymbol{\theta}_{t+1} - \boldsymbol{\theta}^*\right)\right|\Big|\boldsymbol{\theta}_t\right] &\geq \left|1 - \eta\lambda_{\boldsymbol{\theta}_t}\right|\left|\boldsymbol{q}^T(\boldsymbol{\theta}_t - \boldsymbol{\theta}^*)\right| - \eta\delta\left\|\boldsymbol{\theta}_t - \boldsymbol{\theta}^*\right\| - \eta\gamma \\
&\geq \min_{\boldsymbol{\theta}\in\mathcal{A}}\left\{\left|1 - \eta\lambda_{\boldsymbol{\theta}}\right|\right\}\left|\boldsymbol{q}^T(\boldsymbol{\theta}_t - \boldsymbol{\theta}^*)\right| - \eta\delta\left\|\boldsymbol{\theta}_t - \boldsymbol{\theta}^*\right\| - \eta\gamma. \tag{S14}
\end{aligned}$$

Therefore,

$$\begin{aligned}
\mathbb{E}\left[\left|\boldsymbol{q}^T\left(\boldsymbol{\theta}_{t+1} - \boldsymbol{\theta}^*\right)\right|\right] &= \mathbb{E}\left[\mathbb{E}\left[\left|\boldsymbol{q}^T\left(\boldsymbol{\theta}_{t+1} - \boldsymbol{\theta}^*\right)\right|\Big|\boldsymbol{\theta}_t\right]\right] \\
&\geq \min_{\boldsymbol{\theta}\in\mathcal{A}}\left\{\left|1 - \eta\lambda_{\boldsymbol{\theta}}\right|\right\}\mathbb{E}\left[\left|\boldsymbol{q}^T(\boldsymbol{\theta}_t - \boldsymbol{\theta}^*)\right|\right] - \eta\delta\mathbb{E}\left[\left\|\boldsymbol{\theta}_t - \boldsymbol{\theta}^*\right\|\right] - \eta\gamma. \tag{S15}
\end{aligned}$$

Let

$$\boldsymbol{\theta}_0 = \varepsilon \boldsymbol{q} + \boldsymbol{\theta}^*, \tag{S16}$$

observe that $\|\boldsymbol{\theta}_0 - \boldsymbol{\theta}^*\| = \varepsilon$ (*i.e.* $\boldsymbol{\theta}_0 \in \mathcal{B}_\varepsilon(\boldsymbol{\theta}^*)$). Recall that by assumption,

$$\lambda_{\boldsymbol{\theta}} > \frac{2}{\eta} + \delta + \frac{\gamma}{\varepsilon} \tag{S17}$$

for all $\boldsymbol{\theta} \in \mathcal{A}$. Since there is a finite number of modes in $\mathcal{A}$, there exists some $\xi > 0$ such that

$$\lambda_{\boldsymbol{\theta}} \geq \frac{2}{\eta} + \delta + \frac{\gamma}{\varepsilon} + \frac{\xi}{\eta}. \tag{S18}$$

Note that in particular, this implies that $\eta \lambda_{\boldsymbol{\theta}} > 1$, so that

$$\min_{\boldsymbol{\theta} \in \mathcal{A}} \{|1 - \eta \lambda_{\boldsymbol{\theta}}|\} = \min_{\boldsymbol{\theta} \in \mathcal{A}} \{\eta \lambda_{\boldsymbol{\theta}} - 1\} \geq 1 + \eta \delta + \frac{\eta \gamma}{\varepsilon} + \xi. \tag{S19}$$

We now use this to prove the two cases of the lemma.

## III.1 Strongly stable

In this section we show that $\boldsymbol{\theta}^*$ is not strongly stable. From (S15) and (S16) we have that

$$\mathbb{E}\left[\left|\boldsymbol{q}^T (\boldsymbol{\theta}_1 - \boldsymbol{\theta}^*)\right|\right] \geq \min_{\boldsymbol{\theta} \in \mathcal{A}} \{|1 - \eta \lambda_{\boldsymbol{\theta}}|\} \varepsilon - \eta \delta \varepsilon - \eta \gamma. \tag{S20}$$

Note that according to (S19)

$$\min_{\boldsymbol{\theta} \in \mathcal{A}} \{|1 - \eta \lambda_{\boldsymbol{\theta}}|\} \varepsilon - \eta \delta \varepsilon - \eta \gamma \geq \left(1 + \eta \delta + \frac{\eta \gamma}{\varepsilon} + \xi\right) \varepsilon - \eta \delta \varepsilon - \eta \gamma$$
$$= \varepsilon + \xi \varepsilon. \tag{S21}$$

Therefore,

$$\mathbb{E}\left[\|\boldsymbol{\theta}_1 - \boldsymbol{\theta}^*\|\right] \geq \mathbb{E}\left[\left|\boldsymbol{q}^T (\boldsymbol{\theta}_1 - \boldsymbol{\theta}^*)\right|\right] > \varepsilon, \tag{S22}$$

which means that $\boldsymbol{\theta}^*$ is not strongly stable.

## III.2 Stable

In this section we show that if $\delta = 0$ then $\boldsymbol{\theta}^*$ is not stable. Assume that $\delta = 0$, and denote $x_t = \mathbb{E}\left[\left|\boldsymbol{q}^T (\boldsymbol{\theta}_t - \boldsymbol{\theta}^*)\right|\right]$. Then from (S15) we have that

$$x_{t+1} \geq \min_{\boldsymbol{\theta} \in \mathcal{A}} \{|1 - \eta \lambda_{\boldsymbol{\theta}}|\} x_t - \eta \gamma. \tag{S23}$$

Since $\min_{\boldsymbol{\theta} \in \mathcal{A}} \{|1 - \eta \lambda_{\boldsymbol{\theta}}|\} > 1 + \eta \gamma / \varepsilon + \xi$ (see (S19)), we have that

$$x_{t+1} \geq \left(1 + \frac{\eta \gamma}{\varepsilon} + \xi\right) x_t - \eta \gamma. \tag{S24}$$

Observe that from (S16) $x_0 = \varepsilon$. Therefore, from (S24), $x_1 \geq (1 + \xi)\varepsilon > \varepsilon$. Let us show that $x_t > \varepsilon$ also for all $t \geq 1$.

*Proof.* Proof by induction.

**Base.** For $t = 1$ we have $x_1 > \varepsilon$.

**Assumption.** Assume that $x_t > \varepsilon$ holds for some $t \in \mathbb{N}$.

**Step.** Observe that

$$
\begin{aligned}
x_{t+1} &\geq \left(1 + \frac{\eta\gamma}{\varepsilon} + \xi\right) x_t - \eta\gamma \\
&= \left(1 + \xi + \frac{\eta\gamma}{\varepsilon} - \frac{\eta\gamma}{x_t}\right) x_t \\
&> (1 + \xi) x_t \\
&> (1 + \xi)\varepsilon \\
&> \varepsilon,
\end{aligned}
\tag{S25}
$$

where the first inequality is due to (S24), and in the second and third we used the induction's assumption. □

Since $\boldsymbol{q}$ is a unit-norm vector, we have that $\mathbb{E}[\|\boldsymbol{\theta}_t - \boldsymbol{\theta}^*\|] \geq \mathbb{E}\left[\left|\boldsymbol{q}^T(\boldsymbol{\theta}_t - \boldsymbol{\theta}^*)\right|\right] = x_t > \varepsilon$ for all $t \geq 1$. Therefore, $\boldsymbol{\theta}^*$ is not a linearly stable minimum of $\mathcal{L}$, which completes the proof.

We note that $x_t$ is not only larger than $\varepsilon$; it in fact diverges. Indeed, due to the third line of (S25), we have that $x_t > (1 + \xi)^t x_0 \xrightarrow[t\to\infty]{} \infty$.

# IV    Proof of Lemma 3

With slight abuse of notation, we interchangeably use $\boldsymbol{H_\theta}$ and $\boldsymbol{H}_m$, where the meaning in the former is that $\boldsymbol{H_\theta} = \boldsymbol{H}_m$ if $\boldsymbol{\theta} \in \mathcal{S}_m$. We define $\boldsymbol{g}_m = \nabla\psi_m(\boldsymbol{\theta}^*)$ and also occasionally use the notation $\boldsymbol{g_\theta}$ to denote $\boldsymbol{g_\theta} = \boldsymbol{g}_m$ if $\boldsymbol{\theta} \in \mathcal{S}_m$.

The generalized linear dynamical system of GD in this case is

$$
\boldsymbol{\theta}_{t+1} = \boldsymbol{\theta}_t - \eta\left(\boldsymbol{g}_{\boldsymbol{\theta}_t} + \boldsymbol{H}_{\boldsymbol{\theta}_t}(\boldsymbol{\theta}_t - \boldsymbol{\theta}^*)\right).
\tag{S26}
$$

The assumption that $\mathcal{L}$ is differentiable at $\boldsymbol{\theta}^*$ translates to $\boldsymbol{g_\theta} = \boldsymbol{0}$ for all $\boldsymbol{\theta} \in \mathcal{A}$. Here we assume that $\lambda_{\max}(\boldsymbol{H}_m) \leq 2/\eta$ for all $m \in \mathcal{I}$. This means that for all $\boldsymbol{\theta} \in \mathcal{B}_\varepsilon(\boldsymbol{\theta}^*)$ we have that $\lambda_{\max}(\boldsymbol{H_\theta}) \leq 2/\eta$. Thus, considering the eigenvalues of $\boldsymbol{I} - \eta\boldsymbol{H_\theta}$, we have that

$$
\lambda_i(\boldsymbol{I} - \eta\boldsymbol{H_\theta}) = 1 - \eta\lambda_i(\boldsymbol{H_\theta}).
\tag{S27}
$$

Since $0 \leq \eta\lambda_i(\boldsymbol{H_\theta}) \leq \eta\lambda_{\max}(\boldsymbol{H_\theta}) \leq 2$, then

$$
|\lambda_i(\boldsymbol{I} - \eta\boldsymbol{H_\theta})| \leq 1,
\tag{S28}
$$

for all $\boldsymbol{\theta} \in \mathcal{B}_\varepsilon(\boldsymbol{\theta}^*)$. Now, let $\boldsymbol{\theta}_0 \in \mathcal{B}_\varepsilon(\boldsymbol{\theta}^*)$, and let us demonstrate that $\|\boldsymbol{\theta}_t - \boldsymbol{\theta}^*\| \leq \varepsilon$ for all $t > 0$.

*Proof.* Proof by induction.

**Base.** For $t = 0$ we are given that $\|\boldsymbol{\theta}_0 - \boldsymbol{\theta}^*\| \leq \varepsilon$.

**Assumption.** Assume that $\|\boldsymbol{\theta}_t - \boldsymbol{\theta}^*\| \leq \varepsilon$ holds for some $t > 0$.

**Step.** Here we assume that $\boldsymbol{\theta}^*$ is a differentiable minimum, so that for all $\boldsymbol{\theta} \in \mathcal{B}_\varepsilon(\boldsymbol{\theta}^*)$ we have that $\boldsymbol{g_\theta} = \boldsymbol{0}$, and in particular for $\boldsymbol{\theta}_t$. Therefore, $\boldsymbol{\theta}_{t+1}$ satisfies

$$
\boldsymbol{\theta}_{t+1} - \boldsymbol{\theta}^* = (\boldsymbol{I} - \eta\boldsymbol{H}_{\boldsymbol{\theta}_t})(\boldsymbol{\theta}_t - \boldsymbol{\theta}^*).
\tag{S29}
$$

Thus,

$$
\begin{aligned}
\|\boldsymbol{\theta}_{t+1} - \boldsymbol{\theta}^*\| &= \|(\boldsymbol{I} - \eta\boldsymbol{H}_{\boldsymbol{\theta}_t})(\boldsymbol{\theta}_t - \boldsymbol{\theta}^*)\| \\
&\leq \|\boldsymbol{I} - \eta\boldsymbol{H}_{\boldsymbol{\theta}_t}\|_{\mathrm{op}} \|\boldsymbol{\theta}_t - \boldsymbol{\theta}^*\| \\
&= \max_i |\lambda_i(\boldsymbol{I} - \eta\boldsymbol{H_\theta})| \|\boldsymbol{\theta}_t - \boldsymbol{\theta}^*\|.
\end{aligned}
\tag{S30}
$$

Since $\boldsymbol{\theta}_t \in \mathcal{B}_\varepsilon(\boldsymbol{\theta}^*)$, we have $\max_i |\lambda_i(\boldsymbol{I} - \eta\boldsymbol{H_\theta})| \leq 1$ and $\|\boldsymbol{\theta}_t - \boldsymbol{\theta}^*\| \leq \varepsilon$, concluding that $\|\boldsymbol{\theta}_{t+1} - \boldsymbol{\theta}^*\| \leq \varepsilon$. □

# V The gradient of $f$ w.r.t. $\boldsymbol{\theta}$

Let $\boldsymbol{w}^{(1)} = [w_1^{(1)}, \ldots, w_k^{(1)}]^T, \boldsymbol{b}^{(1)} = [b_1^{(1)}, \ldots, b_k^{(1)}]^T \in \mathbb{R}^k$ be the weights and biases of the first layer, and let $\boldsymbol{w}^{(2)} = [w_1^{(2)}, \ldots, w_k^{(2)}]^T \in \mathbb{R}^k, b^{(2)} \in \mathbb{R}$ be the weights and bias of the second layer. Observe that the parameter vector of the network (20) can be expressed as

$$\boldsymbol{\theta} = \left[ \left( \boldsymbol{w}^{(1)} \right)^T, \left( \boldsymbol{b}^{(1)} \right)^T, \left( \boldsymbol{w}^{(2)} \right)^T, b^{(2)} \right]^T \in \mathbb{R}^{3k+1}. \tag{S31}$$

Let $\mathbb{I} : \mathbb{R} \times \mathbb{R}^{3k+1} \mapsto \{0, 1\}^k$ be the activation pattern of all neurons for input $x$, namely $[\mathbb{I}(x; \boldsymbol{\theta})]_i = 1$ if $w_i^{(1)} x + b_i^{(1)} > 0$ and $[\mathbb{I}(x; \boldsymbol{\theta})]_i = 0$ otherwise. Consider the representation of $f$ in Sec. 3, then the gradient of $f$ w.r.t. $\boldsymbol{\theta}$ is given by

$$\nabla_{\boldsymbol{\theta}} f(x) = \begin{pmatrix} \nabla_{\boldsymbol{w}^{(1)}} f(x) \\ \nabla_{\boldsymbol{b}^{(1)}} f(x) \\ \nabla_{\boldsymbol{w}^{(2)}} f(x) \\ \frac{\partial}{\partial b^{(2)}} f(x) \end{pmatrix} = \begin{pmatrix} x \boldsymbol{w}^{(2)} \odot \mathbb{I}(x; \boldsymbol{\theta}) \\ \boldsymbol{w}^{(2)} \odot \mathbb{I}(x; \boldsymbol{\theta}) \\ \left( x \boldsymbol{w}^{(1)} + \boldsymbol{b}^{(1)} \right) \odot \mathbb{I}(x; \boldsymbol{\theta}) \\ 1 \end{pmatrix}, \tag{S32}$$

where $\odot$ denotes the Hadamard product.

# VI Proof of Lemma 4

Recall that $\boldsymbol{\Phi} = [\nabla_{\boldsymbol{\theta}} f(x_1), \nabla_{\boldsymbol{\theta}} f(x_2), \ldots, \nabla_{\boldsymbol{\theta}} f(x_n)]$. In Appendix V we derived an explicit expression for each $\nabla_{\boldsymbol{\theta}} f(x_j)$. Using this expression, and denoting $[\mathbb{I}(x_j; \boldsymbol{\theta}^*)]_i = I_{j,i}$, we have that

$$\max_{\boldsymbol{u} \in \mathbb{S}^{n-1}} \frac{1}{n} \| \boldsymbol{\Phi} \boldsymbol{u} \|^2 \geq \frac{1}{n^2} \| \boldsymbol{\Phi} \mathbf{1} \|^2$$

$$= 1 + \frac{1}{n^2} \sum_{i=1}^{k} \left[ \left( \sum_{j=1}^{n} x_j I_{j,i} w_i^{(2)} \right)^2 + \left( \sum_{j=1}^{n} I_{j,i} w_i^{(2)} \right)^2 + \left( \sum_{j=1}^{n} \sigma \left( w_i^{(1)} x_j + b_i^{(1)} \right) \right)^2 \right]$$

$$= 1 + \frac{1}{n^2} \sum_{i=1}^{k} \left[ \left( w_i^{(2)} \right)^2 \left( \left( \sum_{j=1}^{n} x_j I_{j,i} \right)^2 + \left( \sum_{j=1}^{n} I_{j,i} \right)^2 \right) + \left( \sum_{j=1}^{n} \sigma \left( w_i^{(1)} x_j + b_i^{(1)} \right) \right)^2 \right]$$

$$\geq 1 + \frac{2}{n^2} \sum_{i=1}^{k} \left| w_i^{(2)} \right| \sqrt{ \left( \sum_{j=1}^{n} x_j I_{j,i} \right)^2 + \left( \sum_{j=1}^{n} I_{j,i} \right)^2 } \left| \sum_{j=1}^{n} \sigma \left( w_i^{(1)} x_j + b_i^{(1)} \right) \right|, \tag{S33}$$

where in the first inequality we chose $\boldsymbol{u} = \mathbf{1}/\sqrt{n}$, and in the last step we used $\alpha^2 + \beta^2 \geq 2|\alpha\beta|$. Let $C_i \subseteq \{x_j\}$ be the set of training points for which the $i$th neuron is active, i.e. $C_i = \{x_j : I_{j,i} = 1\}$, and denote $n_i = |C_i|$, that is

$$n_i = \sum_{j=1}^{n} I_{j,i}. \tag{S34}$$

Then,

$$\lambda_{\max}(\nabla_{\boldsymbol{\theta}}^2 \mathcal{L}) \geq 1 + \frac{2}{n^2} \sum_{i=1}^{k} \left| w_i^{(2)} \right| \sqrt{ \left( \sum_{x \in C_i} x \right)^2 + n_i^2 } \left| \sum_{x \in C_i} \left( w_i^{(1)} x + b_i^{(1)} \right) \right|$$

$$= 1 + 2 \sum_{i=1}^{k} \left| w_i^{(2)} \right| \left( \frac{n_i}{n} \right)^2 \sqrt{ \left( \frac{1}{n_i} \sum_{x \in C_i} x \right)^2 + 1 } \left| \frac{1}{n_i} \sum_{x \in C_i} \left( w_i^{(1)} x + b_i^{(1)} \right) \right|$$

$$= 1 + 2 \sum_{i=1}^{k} \left| w_i^{(2)} \right| (\mathbb{P}(X \in C_i))^2 \sqrt{ (\mathbb{E}[X | X \in C_i])^2 + 1 } \left| \mathbb{E} \left[ w_i^{(1)} X + b_i^{(1)} \Big| X \in C_i \right] \right|, \tag{S35}$$

where $X$ is a random sample from the dataset under the uniform distribution. The $i$th summand should be interpreted as 0 if $n_i = 0$. Now, define

$$\tau_i = \begin{cases} -\dfrac{b_i^{(1)}}{w_i^{(1)}}, & w_i^{(1)} \neq 0, \\ 0, & w_i^{(1)} = 0. \end{cases} \tag{S36}$$

We can therefore write

$$1 + 2\sum_{i=1}^{k} \left| w_i^{(2)} \right| (\mathbb{P}(X \in C_i))^2 \sqrt{(\mathbb{E}\left[X | X \in C_i\right])^2 + 1} \left| \mathbb{E}\left[ w_i^{(1)} X + b_i^{(1)} \Big| X \in C_i \right] \right| \geq$$

$$1 + 2\sum_{i=1}^{k} \left| w_i^{(1)} w_i^{(2)} \right| (\mathbb{P}(X \in C_i))^2 \sqrt{(\mathbb{E}\left[X | X \in C_i\right])^2 + 1} \left| \mathbb{E}\left[X - \tau_i | X \in C_i\right] \right| \tag{S37}$$

where the inequality is due to neurons whose $w^{(1)}$ weight is zero. It is easy to see that $\{\tau_i\}$ are in fact the activation thresholds of the neurons. Notice that each neuron can be open either towards the positive direction of the $x$ axis or the negative, therefore $C_i = \{X > \tau_i\}$ or $C_i = \{X < \tau_i\}$. To further bound the top eigenvalue from below, for every neuron we take the direction which minimizes the bound. To this end, we define

$$g^+(x) = \mathbb{P}^2\left(X > x\right) \mathbb{E}\left[X - x | X > x\right] \sqrt{1 + (\mathbb{E}[X | X > x])^2}, \tag{S38}$$

for the positive direction, and

$$g^-(x) = \mathbb{P}^2\left(X < x\right) \mathbb{E}\left[x - X | X < x\right] \sqrt{1 + (\mathbb{E}[X | X < x])^2} \tag{S39}$$

for the negative direction. Therefore, for all $i \in [k]$ we have

$$(\mathbb{P}(X \in C_i))^2 \sqrt{(\mathbb{E}\left[X | X \in C_i\right])^2 + 1} \left| \mathbb{E}\left[X - \tau_i | X \in C_i\right] \right| \geq \min\left\{g^+(\tau_i), g^-(\tau_i)\right\}. \tag{S40}$$

Thus, since $\boldsymbol{\theta} \in \Omega(f)$ we have

$$\lambda_{\max}(\nabla_{\boldsymbol{\theta}}^2 \mathcal{L}) \geq 1 + 2\sum_{i=1}^{k} \left| w_i^{(1)} w_i^{(2)} \right| \min\left\{g^+(\tau_i), g^-(\tau_i)\right\}$$

$$\geq 1 + 2\int_{x_{\min}}^{x_{\max}} |f''(x)| \min\left\{g^+(x), g^-(x)\right\} \mathrm{d}x, \tag{S41}$$

where we used the fact that[1] $f''(x) = \sum_{i=1}^{k} w_i^{(1)} w_i^{(2)} \delta(x - \tau_i)$. The last inequality in (S41) is due to scenarios in which multiple neurons become active at the same threshold. In such a case, the upper expression adds their weights in absolute value, while the lower expression adds their signed weights and thus cannot be greater. Overall, we reached a bound that is implementation free, *i.e.* does not depend on $\boldsymbol{\theta}$, which holds for all $\boldsymbol{\theta} \in \Omega(f)$. Therefore, it also holds for the implementation of $f$ corresponding to the flattest minimum, namely

$$\min_{\boldsymbol{\theta} \in \Omega(f)} \lambda_{\max}(\nabla_{\boldsymbol{\theta}}^2 \mathcal{L}) \geq 1 + 2\int_{-\infty}^{\infty} |f''(x)| \, g(x) \mathrm{d}x. \tag{S42}$$

## VII    Switching system formulation for single hidden layer ReLU networks

In this section we apply the switching system framework to one-hidden-layer networks trained with quadratic loss. Before we present the loss $\mathcal{L}$ as resulting from a discrete set of analytic functions, let us first present the predictor function $f(x; \boldsymbol{\theta})$ in such a way.

Note that the function implemented by the network can be written in vector form as

$$f(x; \boldsymbol{\theta}) = \left(\boldsymbol{w}^{(2)}\right)^T \sigma\left(x\boldsymbol{w}^{(1)} + \boldsymbol{b}^{(1)}\right) + b^{(2)}, \tag{S43}$$

---

[1]$\delta$ denotes the Dirac delta function.

where $\sigma(\cdot)$ denotes the element-wise ReLU function. Let $\mathbb{I} : \mathbb{R} \times \mathbb{R}^{3k+1} \mapsto \{0, 1\}^k$ be the activation pattern of all neurons for input $x$, namely $[\mathbb{I}(x; \boldsymbol{\theta})]_i = 1$ if $w_i^{(1)} x + b_i^{(1)} > 0$ and $[\mathbb{I}(x; \boldsymbol{\theta})]_i = 0$ otherwise. Then, we can also write $f(x; \boldsymbol{\theta})$ as

$$f(x; \boldsymbol{\theta}) = \left( \mathbb{I}(x; \boldsymbol{\theta}) \odot \boldsymbol{w}^{(2)} \right)^T \left( x \boldsymbol{w}^{(1)} + \boldsymbol{b}^{(1)} \right) + b^{(2)}, \tag{S44}$$

where $\odot$ denotes the Hadamard product. Let $\mathcal{Z} = \{ \mathbb{I}^m \}_{m=1}^{2^k}$ be the set of all $k$-dimensional binary patterns $\{0, 1\}^k$, and define

$$\phi(x; \boldsymbol{\theta}, m) = \left( \mathbb{I}^m \odot \boldsymbol{w}^{(2)} \right)^T \left( x \boldsymbol{w}^{(1)} + \boldsymbol{b}^{(1)} \right) + b^{(2)} \tag{S45}$$

so that $f(x; \boldsymbol{\theta})$ can be expressed as

$$f(x; \boldsymbol{\theta}) = \phi(x; \boldsymbol{\theta}, m) \quad \text{if} \quad \mathbb{I}(x; \boldsymbol{\theta}) = \mathbb{I}^m. \tag{S46}$$

Observe that

$$\phi(x; \boldsymbol{\theta}^*, m) = \phi(x; \boldsymbol{\theta}^*, m') \tag{S47}$$

if $\mathbb{I}^m$ and $\mathbb{I}^{m'}$ differ in indices at which the vector $x \boldsymbol{w}^{(1)} + \boldsymbol{b}^{(1)}$ has zeros.

We next identify the modes of the loss. Recall that the stochastic loss function is given by

$$\hat{\mathcal{L}}_t(\boldsymbol{\theta}) = \frac{1}{2B} \sum_{j \in \mathfrak{B}_t} \left( f(x_j; \boldsymbol{\theta}) - y_j \right)^2. \tag{S48}$$

Let $\mathcal{Z}^n = \mathcal{Z} \times \cdots \times \mathcal{Z}$ ($n$ times) be the set of $2^{nk}$ possible $n$-tuples of $k$-dimensional binary vectors, indexed by the multi-index $\boldsymbol{m} = (m_1, \ldots, m_n)$, where $m_j \in \{1, \ldots, 2^k\}$ for every $j$. Define

$$\hat{\psi}_{\boldsymbol{m}}^{(t)}(\boldsymbol{\theta}) = \frac{1}{2B} \sum_{j \in \mathfrak{B}_t} \left( \phi(x_j; \boldsymbol{\theta}, m_j) - y_j \right)^2, \tag{S49}$$

and

$$\mathcal{S}_{\boldsymbol{m}} = \left\{ \boldsymbol{\theta} \,\middle|\, \mathbb{I}(x_j; \boldsymbol{\theta}) = \mathbb{I}_{m_j} \,\forall j \in [n] \right\}. \tag{S50}$$

Then the loss $\hat{\mathcal{L}}_t(\boldsymbol{\theta})$ can be expressed as

$$\hat{\mathcal{L}}_t(\boldsymbol{\theta}) = \hat{\psi}_{\boldsymbol{m}}^{(t)}(\boldsymbol{\theta}) \quad \text{if} \quad \boldsymbol{\theta} \in \mathcal{S}_{\boldsymbol{m}}. \tag{S51}$$

Differentiating $\hat{\psi}_{\boldsymbol{m}}^{(t)}$ w.r.t. $\boldsymbol{\theta}$ we have

$$\nabla_{\boldsymbol{\theta}} \hat{\psi}_{\boldsymbol{m}}^{(t)}(\boldsymbol{\theta}) = \frac{1}{B} \sum_{j \in \mathfrak{B}_t} \left( \phi(x_j; \boldsymbol{\theta}, m_j) - y_j \right) \nabla_{\boldsymbol{\theta}} \phi(x_j; \boldsymbol{\theta}, m_j), \tag{S52}$$

where for any $x \in \mathbb{R}$ and $\mathbb{I}^m \in \{0, 1\}^k$ the gradient of $\phi$ is given by

$$\nabla_{\boldsymbol{\theta}} \phi(x; \boldsymbol{\theta}, m) = \begin{pmatrix} \nabla_{\boldsymbol{w}^{(1)}} \phi(x; \boldsymbol{\theta}, \mathbb{I}^m) \\ \nabla_{\boldsymbol{b}^{(1)}} \phi(x; \boldsymbol{\theta}, \mathbb{I}^m) \\ \nabla_{\boldsymbol{w}^{(2)}} \phi(x; \boldsymbol{\theta}, \mathbb{I}^m) \\ \frac{\partial}{\partial b^{(2)}} \phi(x_j; \boldsymbol{\theta}, \mathbb{I}^m) \end{pmatrix} = \begin{pmatrix} x \boldsymbol{w}^{(2)} \odot \mathbb{I}^m \\ \boldsymbol{w}^{(2)} \odot \mathbb{I}^m \\ \left( x \boldsymbol{w}^{(1)} + \boldsymbol{b}^{(1)} \right) \odot \mathbb{I}^m \\ 1 \end{pmatrix}. \tag{S53}$$

Let $\boldsymbol{\theta}^*$ be a global minimum of $\mathcal{L}$ (*i.e.* $f(x_j) = y_j$ for all $j \in [n]$) that is not twice-differentiable. In our case $\mathcal{I}$ is given by[2]

$$\begin{aligned} \mathcal{I} &= \left\{ \boldsymbol{m} : \boldsymbol{\theta}^* \in \bar{\mathcal{S}}_{\boldsymbol{m}} \right\} \\ &= \left\{ \boldsymbol{m} : [\mathbb{I}^{m_j}]_i = [\mathbb{I}(x_j; \boldsymbol{\theta}^*)]_i \quad \text{whenever} \quad w_i^{(1)} x_j + b_i^{(1)} \neq 0, \quad \forall j \in [n] \right\}. \end{aligned} \tag{S54}$$

---

[2] $\bar{\mathcal{S}}_m$ denotes the closure of $\mathcal{S}_m$

Observe that for $\mathbb{I}^{m'_j} = \mathbb{I}(x_j; \boldsymbol{\theta}^*)$ we have that

$$\phi(x_j; \boldsymbol{\theta}^*, m'_j) = f(x; \boldsymbol{\theta}^*) = y_j. \tag{S55}$$

Therefore according to (S47), $\forall \boldsymbol{m} \in \mathcal{I}$ we have that

$$\phi(x_j; \boldsymbol{\theta}^*, m_j) = \phi(x_j; \boldsymbol{\theta}^*, m'_j) = y_j \tag{S56}$$

for all $j \in [n]$. Thus, from (S52), for any $\boldsymbol{m} \in \mathcal{I}$ we have

$$\hat{\boldsymbol{g}}_{\boldsymbol{m}}^{(t)} = \nabla_{\boldsymbol{\theta}} \hat{\psi}_{\boldsymbol{m}}^{(t)}(\boldsymbol{\theta}^*) = \boldsymbol{0}. \tag{S57}$$

Finally, considering full batch ($\mathfrak{B}_t = [n]$), the Hessian of $\psi_{\boldsymbol{m}}$ is given by

$$\nabla_{\boldsymbol{\theta}}^2 \psi_{\boldsymbol{m}}(\boldsymbol{\theta}) = \frac{1}{n} \sum_{j=1}^n \left( \nabla_{\boldsymbol{\theta}} \phi(x_j; \boldsymbol{\theta}, m_j) \right) \left( \nabla_{\boldsymbol{\theta}} \phi(x_j; \boldsymbol{\theta}, m_j) \right)^T$$

$$+ \frac{1}{n} \sum_{j=1}^n \left( \phi(x_j; \boldsymbol{\theta}, m_j) - y_j \right) \nabla_{\boldsymbol{\theta}}^2 \phi(x_j; \boldsymbol{\theta}, m_j). \tag{S58}$$

Calculating this Hessian at $\boldsymbol{\theta}^*$ for $\boldsymbol{m} \in \mathcal{I}$ we get

$$\boldsymbol{H}_{\boldsymbol{m}} = \nabla_{\boldsymbol{\theta}}^2 \hat{\psi}_{\boldsymbol{m}}(\boldsymbol{\theta}^*) = \frac{1}{n} \sum_{j=1}^n \left( \nabla_{\boldsymbol{\theta}} \phi(x_j; \boldsymbol{\theta}^*, m_j) \right) \left( \nabla_{\boldsymbol{\theta}} \phi(x_j; \boldsymbol{\theta}^*, m_j) \right)^T. \tag{S59}$$

Let us denote the tangent features matrix of the $\boldsymbol{m}$'th mode by

$$\boldsymbol{\Phi}_{\boldsymbol{m}} = [\nabla_{\boldsymbol{\theta}} \phi(x_1; \boldsymbol{\theta}^*, m_1), \nabla_{\boldsymbol{\theta}} \phi(x_2; \boldsymbol{\theta}^*, m_2), \dots, \nabla_{\boldsymbol{\theta}} \phi(x_n; \boldsymbol{\theta}^*, m_n)]. \tag{S60}$$

Then the Hessian at $\boldsymbol{\theta}^*$ of $\psi_{\boldsymbol{m}}$ for $\boldsymbol{m} \in \mathcal{I}$ can be expressed as $\nabla_{\boldsymbol{\theta}}^2 \mathcal{L} = \boldsymbol{\Phi}_{\boldsymbol{m}} \boldsymbol{\Phi}_{\boldsymbol{m}}^T / n$.

## VIII Proof of Lemma 5

In this section we use notations from Appendix VII. Observe that $\mathcal{H} = \{\boldsymbol{H}_{\boldsymbol{m}} : \boldsymbol{m} \in \mathcal{I}\}$. Let $\boldsymbol{H}, \tilde{\boldsymbol{H}} \in \mathcal{H}$ with corresponding indices $\boldsymbol{m}, \tilde{\boldsymbol{m}} \in \mathcal{I}$, namely $\boldsymbol{H} = \boldsymbol{H}_{\boldsymbol{m}}$ and $\tilde{\boldsymbol{H}} = \boldsymbol{H}_{\tilde{\boldsymbol{m}}}$. Denote the top eigenpair of $\boldsymbol{H}_{\boldsymbol{m}}$ by $\lambda \geq 0$ and $\boldsymbol{q} \in \mathbb{S}^{3k}$. Here, we are interested in an upper bound for $\|\boldsymbol{H}_{\tilde{\boldsymbol{m}}} \boldsymbol{q} - \lambda \boldsymbol{q}\|$. Note that

$$\|\boldsymbol{H}_{\tilde{\boldsymbol{m}}} \boldsymbol{q} - \lambda \boldsymbol{q}\| = \|\boldsymbol{H}_{\tilde{\boldsymbol{m}}} \boldsymbol{q} - \boldsymbol{H}_{\boldsymbol{m}} \boldsymbol{q}\| = \|(\boldsymbol{H}_{\tilde{\boldsymbol{m}}} - \boldsymbol{H}_{\boldsymbol{m}}) \boldsymbol{q}\|$$
$$\leq \|\boldsymbol{H}_{\tilde{\boldsymbol{m}}} - \boldsymbol{H}_{\boldsymbol{m}}\|_{\text{op}} \|\boldsymbol{q}\| = \|\boldsymbol{H}_{\tilde{\boldsymbol{m}}} - \boldsymbol{H}_{\boldsymbol{m}}\|_{\text{op}}. \tag{S61}$$

Let $\boldsymbol{\Phi}_{\boldsymbol{m}}$ and $\boldsymbol{\Phi}_{\tilde{\boldsymbol{m}}}$ be the corresponding tangent matrices of $\boldsymbol{H}_{\boldsymbol{m}}$ and $\boldsymbol{H}_{\tilde{\boldsymbol{m}}}$, i.e.

$$\boldsymbol{H}_{\boldsymbol{m}} = \frac{1}{n} \boldsymbol{\Phi}_{\boldsymbol{m}} \boldsymbol{\Phi}_{\boldsymbol{m}}^T \quad \text{and} \quad \boldsymbol{H}_{\tilde{\boldsymbol{m}}} = \frac{1}{n} \boldsymbol{\Phi}_{\tilde{\boldsymbol{m}}} \boldsymbol{\Phi}_{\tilde{\boldsymbol{m}}}^T. \tag{S62}$$

Then, using the submultiplicativity and subadditivity of the operator norm we have

$$\|\boldsymbol{H}_{\boldsymbol{m}} - \boldsymbol{H}_{\tilde{\boldsymbol{m}}}\|_{\text{op}} = \frac{1}{n} \left\| \boldsymbol{\Phi}_{\boldsymbol{m}} \boldsymbol{\Phi}_{\boldsymbol{m}}^T - \boldsymbol{\Phi}_{\tilde{\boldsymbol{m}}} \boldsymbol{\Phi}_{\tilde{\boldsymbol{m}}}^T \right\|_{\text{op}}$$
$$= \frac{1}{n} \left\| (\boldsymbol{\Phi}_{\boldsymbol{m}} - \boldsymbol{\Phi}_{\tilde{\boldsymbol{m}}}) \boldsymbol{\Phi}_{\boldsymbol{m}}^T - \boldsymbol{\Phi}_{\tilde{\boldsymbol{m}}} (\boldsymbol{\Phi}_{\tilde{\boldsymbol{m}}} - \boldsymbol{\Phi}_{\boldsymbol{m}})^T \right\|_{\text{op}}$$
$$\leq \frac{1}{n} \left\| (\boldsymbol{\Phi}_{\boldsymbol{m}} - \boldsymbol{\Phi}_{\tilde{\boldsymbol{m}}}) \boldsymbol{\Phi}_{\boldsymbol{m}}^T \right\|_{\text{op}} + \left\| \boldsymbol{\Phi}_{\tilde{\boldsymbol{m}}} (\boldsymbol{\Phi}_{\tilde{\boldsymbol{m}}} - \boldsymbol{\Phi}_{\boldsymbol{m}})^T \right\|_{\text{op}}$$
$$\leq \frac{1}{n} \left( \|\boldsymbol{\Phi}_{\boldsymbol{m}} - \boldsymbol{\Phi}_{\tilde{\boldsymbol{m}}}\|_{\text{op}} \|\boldsymbol{\Phi}_{\boldsymbol{m}}\|_{\text{op}} + \|\boldsymbol{\Phi}_{\tilde{\boldsymbol{m}}}\|_{\text{op}} \|\boldsymbol{\Phi}_{\tilde{\boldsymbol{m}}} - \boldsymbol{\Phi}_{\boldsymbol{m}}\|_{\text{op}} \right)$$
$$= \frac{\|\boldsymbol{\Phi}_{\boldsymbol{m}} - \boldsymbol{\Phi}_{\tilde{\boldsymbol{m}}}\|_{\text{op}}}{\sqrt{n}} \left( \frac{\|\boldsymbol{\Phi}_{\boldsymbol{m}}\|_{\text{op}}}{\sqrt{n}} + \frac{\|\boldsymbol{\Phi}_{\tilde{\boldsymbol{m}}}\|_{\text{op}}}{\sqrt{n}} \right)$$
$$= \frac{\|\boldsymbol{\Phi}_{\boldsymbol{m}} - \boldsymbol{\Phi}_{\tilde{\boldsymbol{m}}}\|_{\text{op}}}{\sqrt{n}} \left( \sqrt{\lambda_{\max}(\boldsymbol{H}_{\boldsymbol{m}})} + \sqrt{\lambda_{\max}(\boldsymbol{H}_{\tilde{\boldsymbol{m}}})} \right). \tag{S63}$$

Let us denote by $p_j$ the number of neurons whose activation threshold coincides with $x_j$. This means that the total number of toggling neurons between two modes in $\mathcal{I}$ is at most $p = \sum_{j=1}^{n} p_j$. We denote the maximal number of toggling neurons for a single sample between any two modes in $\mathcal{I}$ as $p_{\max} = \max\{p_j\}$. Additionally, let $a(i,j) = 1$ if the $i$th neuron toggles on $x_j$ from mode $\boldsymbol{m}$ to mode $\tilde{\boldsymbol{m}}$, and $a(i,j) = 0$ otherwise (mathematically, $a(i,j) = |[\mathbb{I}^{m_j} - \mathbb{I}^{\tilde{m}_j}]_i|$). Since we study zero-loss (perfect interpolation) solutions, we assume that $x_i \neq x_j$ for all $i \neq j$ (see Sec. 3). Therefore, a neuron can toggle on one training sample at most. Thus, in our setting we have that

$$\sum_{i=1}^{k} a(i,j) \le p_j, \qquad \text{and} \qquad \sum_{j=1}^{n} a(i,j) \le 1. \tag{S64}$$

Then, using the definition of $\boldsymbol{\Phi_m}$ (see (S60)), we have

$$\|\boldsymbol{\Phi_m} - \boldsymbol{\Phi_{\tilde m}}\|_{\text{op}}^2 = \max_{\boldsymbol{u} \in \mathbb{S}^{n-1}} \|(\boldsymbol{\Phi_m} - \boldsymbol{\Phi_{\tilde m}})\boldsymbol{u}\|^2$$

$$= \max_{\boldsymbol{u} \in \mathbb{S}^{n-1}} \sum_{i=1}^{3k+1} \left( \sum_{j=1}^{n} \boldsymbol{u}_j \left[ \nabla_{\boldsymbol\theta}\phi(x_j; \boldsymbol\theta^*, m_j) - \nabla_{\boldsymbol\theta}\phi(x_j; \boldsymbol\theta^*, \tilde{m}_j) \right]_i \right)^2$$

$$\le \max_{\boldsymbol{u} \in \mathbb{S}^{n-1}} \sum_{i=1}^{3k+1} \left( \sum_{j=1}^{n} |\boldsymbol{u}_j| \left| [\nabla_{\boldsymbol\theta}\phi(x_j; \boldsymbol\theta^*, m_j) - \nabla_{\boldsymbol\theta}\phi(x_j; \boldsymbol\theta^*, \tilde{m}_j)]_i \right| \right)^2. \tag{S65}$$

Using the particular structure of $\nabla_{\boldsymbol\theta}\phi(x; \boldsymbol\theta, m)$ from (S53) we have that

$$\sum_{i=1}^{3k+1} \left( \sum_{j=1}^{n} |\boldsymbol{u}_j| \left| [\nabla_{\boldsymbol\theta}\phi(x_j; \boldsymbol\theta^*, m_j) - \nabla_{\boldsymbol\theta}\phi(x_j; \boldsymbol\theta^*, \tilde{m}_j)]_i \right| \right)^2$$

$$= \sum_{i=1}^{k} \sum_{m=0}^{2} \left( \sum_{j=1}^{n} |\boldsymbol{u}_j| \left| [\nabla_{\boldsymbol\theta}\phi(x_j; \boldsymbol\theta^*, m_j) - \nabla_{\boldsymbol\theta}\phi(x_j; \boldsymbol\theta^*, \tilde{m}_j)]_{km+i} \right| \right)^2$$

$$\le \sum_{i=1}^{k} \sum_{m=0}^{2} \left( \sum_{j=1}^{n} |\boldsymbol{u}_j| C a(i,j) \right)^2$$

$$= 3C^2 \sum_{i=1}^{k} \left( \sum_{j=1}^{n} |\boldsymbol{u}_j| a(i,j) \right)^2, \tag{S66}$$

where we used the assumption that $\|\phi(x_j; \boldsymbol\theta^*, m_j)\|_\infty \le C$ for all $\boldsymbol{m} \in \mathcal{I}$ and $j \in [n]$. Note that the series $\sum_{j=1}^{n} |\boldsymbol{u}_j| a(i,j)$ has at most one term, so that for all $\boldsymbol{u} \in \mathbb{S}^{n-1}$ we have

$$\sum_{i=1}^{k} \left( \sum_{j=1}^{n} |\boldsymbol{u}_j| a(i,j) \right)^2 = \sum_{i=1}^{k} \sum_{j=1}^{n} \boldsymbol{u}_j^2 a(i,j)$$

$$= \sum_{j=1}^{n} \boldsymbol{u}_j^2 \sum_{i=1}^{k} a(i,j)$$

$$\le \sum_{j=1}^{n} \boldsymbol{u}_j^2 p_j$$

$$\le p_{\max} \sum_{j=1}^{n} \boldsymbol{u}_j^2$$

$$= p_{\max}, \tag{S67}$$

where in the first inequality we used (S64). Thus,

$$\|\boldsymbol{\Phi_m} - \boldsymbol{\Phi_{\tilde m}}\|_{\text{op}} \le \sqrt{3} C \sqrt{p_{\max}}. \tag{S68}$$

Substituting this bound in (S63) results in

$$\|\boldsymbol{H}_{\tilde{\boldsymbol{m}}}\boldsymbol{q} - \lambda\boldsymbol{q}\| \le \sqrt{3}C\left(\sqrt{\lambda_{\max}(\boldsymbol{H}_{\boldsymbol{m}})} + \sqrt{\lambda_{\max}(\boldsymbol{H}_{\tilde{\boldsymbol{m}}})}\right)\sqrt{\frac{p_{\max}}{n}}, \tag{S69}$$

or, with a simpler notation,

$$\left\|\tilde{\boldsymbol{H}}\boldsymbol{q} - \lambda_{\max}(\boldsymbol{H})\boldsymbol{q}\right\| \le \sqrt{3}C\left(\sqrt{\lambda_{\max}(\boldsymbol{H})} + \sqrt{\lambda_{\max}(\tilde{\boldsymbol{H}})}\right)\sqrt{\frac{p_{\max}}{n}}, \tag{S70}$$

which completes the proof.

## IX  Proof parts for Theorem 2

Applying Lemma 2 with $\gamma = 0$, $\delta = 2\sqrt{3}C\sqrt{\lambda_{\max}^{\text{upper}}}\sqrt{\frac{p_{\max}}{n}}$ and $\lambda_{\boldsymbol{m}} = \lambda_{\max}^{\text{upper}}$ for all $\boldsymbol{m} \in \mathcal{I}$ (such that $\lambda^{\text{lower}} = \lambda_{\max}^{\text{upper}}$), we get that if $\boldsymbol{\theta}^*$ is strongly stable then

$$\lambda_{\max}^{\text{upper}}(\boldsymbol{\theta}^*) \le \frac{2}{\eta} + 2\sqrt{3}C\sqrt{\lambda_{\max}^{\text{upper}}(\boldsymbol{\theta}^*)}\sqrt{\frac{p_{\max}}{n}}. \tag{S71}$$

Let us denote $\zeta^2 = \lambda_{\max}^{\text{upper}}(\boldsymbol{\theta}^*)$ for $\zeta \ge 0$, $\alpha = 2\sqrt{3}C\sqrt{\frac{p_{\max}}{n}}$ and $\beta = \frac{2}{\eta}$. Thus (S71) becomes

$$\zeta^2 - \alpha\zeta - \beta \le 0 \qquad \text{and} \qquad \zeta \ge 0. \tag{S72}$$

The roots of the left hand side parabola are given by

$$\zeta_{1,2} = \frac{\alpha \pm \sqrt{\alpha^2 + 4\beta}}{2}. \tag{S73}$$

Hence (S72) is equivalent to

$$0 \le \zeta \le \frac{\alpha + \sqrt{\alpha^2 + 4\beta}}{2}. \tag{S74}$$

Therefore,

$$\begin{aligned}
\lambda_{\max}^{\text{upper}}(\boldsymbol{\theta}^*) &= \zeta^2 \\
&\le \left(\frac{\alpha + \sqrt{\alpha^2 + 4\beta}}{2}\right)^2 \\
&= \frac{1}{4}\left(2\alpha^2 + 4\beta + 2\alpha\sqrt{\alpha^2 + 4\beta}\right) \\
&= \frac{2}{\eta} + 3C^2\frac{p_{\max}}{n} + C\sqrt{3\frac{p_{\max}}{n}\left(3C^2\frac{p_{\max}}{n} + \frac{2}{\eta}\right)}.
\end{aligned} \tag{S75}$$

## X  Generalization of Lemma 4 to global minima that are not twice-differentiable

In Appendix VII we give explicit expressions for the Hessian matrices and gradient vectors of the switching system corresponding to the generalized linear dynamics of SGD near global minima in our setting. Specifically, we use a multi-index $\boldsymbol{m} = (m_1, m_2, \ldots, m_n)$ to enumerate all modes of the switching system. Then, for a global minimum $\boldsymbol{\theta}^*$ that is not twice differentiable, we show that when $\boldsymbol{m} \in \mathcal{I}$, the Hessian $\boldsymbol{H}_{\boldsymbol{m}} = \nabla^2\psi_{\boldsymbol{m}}(\boldsymbol{\theta}^*)$ can be expressed as $\boldsymbol{\Phi}_{\boldsymbol{m}}\boldsymbol{\Phi}_{\boldsymbol{m}}^T/n$, with

$$\boldsymbol{\Phi}_{\boldsymbol{m}} = \begin{bmatrix} x_1\boldsymbol{w}^{(2)} \odot \mathbb{I}^{m_1} & x_2\boldsymbol{w}^{(2)} \odot \mathbb{I}^{m_2} & \cdots & x_n\boldsymbol{w}^{(2)} \odot \mathbb{I}^{m_n} \\ \boldsymbol{w}^{(2)} \odot \mathbb{I}^{m_1} & \boldsymbol{w}^{(2)} \odot \mathbb{I}^{m_2} & \cdots & \boldsymbol{w}^{(2)} \odot \mathbb{I}^{m_n} \\ \left(x_1\boldsymbol{w}^{(1)} + \boldsymbol{b}^{(1)}\right) \odot \mathbb{I}^{m_1} & \left(x_2\boldsymbol{w}^{(1)} + \boldsymbol{b}^{(1)}\right) \odot \mathbb{I}^{m_2} & \cdots & \left(x_n\boldsymbol{w}^{(1)} + \boldsymbol{b}^{(1)}\right) \odot \mathbb{I}^{m_n} \\ 1 & 1 & \cdots & 1 \end{bmatrix}, \tag{S76}$$

where $\{\mathbb{I}^{m_j}\}_{j=1}^n$ is a $n$-tuple of $k$-dimensional binary vectors. In particular, the index $\boldsymbol{m}'$ for which

$$\forall j \in [n] \qquad \mathbb{I}^{m'_j} = \mathbb{I}(x_j; \boldsymbol{\theta}^*) \tag{S77}$$

is also in $\mathcal{I}$ (see Appendix VII for details). Since $\boldsymbol{H}_{\boldsymbol{m}'}$ has that same structure and activation pattern as the Hessian matrices considered in Lemma 4, the proof of Lemma 4 (see Appendix VI) applies for $\boldsymbol{H}_{\boldsymbol{m}'}$ as well. Therefore,

$$\lambda_{\max}^{\text{upper}} = \max_{\boldsymbol{m} \in \mathcal{I}} \{\lambda_{\max}(\boldsymbol{H}_{\boldsymbol{m}})\} \geq \lambda_{\max}(\boldsymbol{H}_{\boldsymbol{m}'}) \geq 1 + 2 \int_{-\infty}^{\infty} |f''(x)| \, g(x) \mathrm{d}x, \tag{S78}$$

where in the last inequality we used Lemma 4. Since this lower bound is implementation independent, we deduce that

$$\min_{\boldsymbol{\theta} \in \Omega(f)} \lambda_{\max}^{\text{upper}}(\boldsymbol{\theta}) \geq 1 + 2 \int_{-\infty}^{\infty} |f''(x)| \, g(x) \mathrm{d}x. \tag{S79}$$

## XI Determining the sharpness of the flattest implementation in experiments

In this section we explain how we determine the sharpness of the flattest implementation. Recall that any solution in $\mathcal{F}$ can be implemented in various ways. Specifically, for each neuron, we can multiply the weight of the first layer by some positive constant, and divide the weight of the second layer by the same constant. This results in different implementations of the same end-to-end function. Mathematically, let $\boldsymbol{\theta}$ be a minimum of the training loss, where

$$\boldsymbol{\theta} = \left[ w_1^{(1)}, \ldots, w_k^{(1)}, b_1^{(1)}, \ldots, b_k^{(1)}, w_1^{(2)}, \ldots, w_k^{(2)}, b^{(2)} \right]^T \in \mathbb{R}^{3k+1}. \tag{S80}$$

Then for any $\boldsymbol{\beta} = [\beta_1, \beta_2, \ldots, \beta_k] \in \mathbb{R}_+^k$ we define

$$\tilde{\boldsymbol{\theta}}(\boldsymbol{\beta}) = \left[ \beta_1 w_1^{(1)}, \ldots, \beta_k w_k^{(1)}, \beta_1 b_1^{(1)}, \ldots, \beta_k b_k^{(1)}, \frac{w_1^{(2)}}{\beta_1}, \ldots, \frac{w_k^{(2)}}{\beta_k}, b^{(2)} \right]^T \in \mathbb{R}^{3k+1}. \tag{S81}$$

Clearly, a network with parameters $\tilde{\boldsymbol{\theta}}(\boldsymbol{\beta})$ implements the same end-to-end function as a network with parameters $\boldsymbol{\theta}$. Thus, to determine the sharpness of the flattest implementation we want to solve the optimization problem

$$\min_{\boldsymbol{\beta} \in \mathbb{R}_+^k} \lambda_{\max}\left(\nabla^2 \mathcal{L}\left(\tilde{\boldsymbol{\theta}}(\boldsymbol{\beta})\right)\right). \tag{S82}$$

In our simulations, we found the minimum of this objective using GD.

To verify that we indeed converged to the global minimum, we used the fact that the objective can be explicitly written as

$$\min_{\boldsymbol{\beta} \in \mathbb{R}_+^k} \max_{\boldsymbol{q} \in \mathbb{S}^{3k}} \boldsymbol{q}^T \nabla^2 \mathcal{L}\left(\tilde{\boldsymbol{\theta}}(\boldsymbol{\beta})\right) \boldsymbol{q}. \tag{S83}$$

From the max–min inequality we know that

$$\min_{\boldsymbol{\beta} \in \mathbb{R}_+^k} \max_{\boldsymbol{q} \in \mathbb{S}^{3k}} \boldsymbol{q}^T \nabla^2 \mathcal{L}\left(\tilde{\boldsymbol{\theta}}(\boldsymbol{\beta})\right) \boldsymbol{q} \geq \max_{\boldsymbol{q} \in \mathbb{S}^{3k}} \min_{\boldsymbol{\beta} \in \mathbb{R}_+^k} \boldsymbol{q}^T \nabla^2 \mathcal{L}\left(\tilde{\boldsymbol{\theta}}(\boldsymbol{\beta})\right) \boldsymbol{q}. \tag{S84}$$

Therefore, for any $\tilde{\boldsymbol{q}} \in \mathbb{S}^{n-1}$ and $\tilde{\boldsymbol{\beta}} \in \mathbb{R}_+^k$ we have that

$$\lambda_{\max}\left(\nabla^2 \mathcal{L}\left(\tilde{\boldsymbol{\theta}}(\tilde{\boldsymbol{\beta}})\right)\right) \geq \min_{\boldsymbol{\beta} \in \mathbb{R}_+^k} \lambda_{\max}\left(\nabla^2 \mathcal{L}\left(\tilde{\boldsymbol{\theta}}(\boldsymbol{\beta})\right)\right)$$

$$\geq \max_{\boldsymbol{q} \in \mathbb{S}^{3k}} \min_{\boldsymbol{\beta} \in \mathbb{R}_+^k} \boldsymbol{q}^T \nabla^2 \mathcal{L}\left(\tilde{\boldsymbol{\theta}}(\boldsymbol{\beta})\right) \boldsymbol{q}$$

$$\geq \min_{\boldsymbol{\beta} \in \mathbb{R}_+^k} \tilde{\boldsymbol{q}}^T \nabla^2 \mathcal{L}\left(\tilde{\boldsymbol{\theta}}(\boldsymbol{\beta})\right) \tilde{\boldsymbol{q}}. \tag{S85}$$

This implies that if for some $\tilde{\boldsymbol{\beta}}$ and $\tilde{\boldsymbol{q}}$ we have that $\lambda_{\max}\left(\nabla^2 \mathcal{L}\left(\tilde{\boldsymbol{\theta}}(\tilde{\boldsymbol{\beta}})\right)\right) = \min_{\boldsymbol{\beta} \in \mathbb{R}_+^k} \tilde{\boldsymbol{q}}^T \nabla^2 \mathcal{L}\left(\tilde{\boldsymbol{\theta}}(\boldsymbol{\beta})\right) \tilde{\boldsymbol{q}}$ then it follows that

$$\lambda_{\max}\left(\nabla^2 \mathcal{L}\left(\tilde{\boldsymbol{\theta}}(\tilde{\boldsymbol{\beta}})\right)\right) = \min_{\boldsymbol{\beta} \in \mathbb{R}_+^k} \lambda_{\max}\left(\nabla^2 \mathcal{L}\left(\tilde{\boldsymbol{\theta}}(\boldsymbol{\beta})\right)\right) = \min_{\boldsymbol{\beta} \in \mathbb{R}_+^k} \tilde{\boldsymbol{q}}^T \nabla^2 \mathcal{L}\left(\tilde{\boldsymbol{\theta}}(\boldsymbol{\beta})\right) \tilde{\boldsymbol{q}}. \tag{S86}$$

Now, as we show below, the right hand side has a simple closed form expression. Namely, in this case we can determine the optimal value of the optimization problem. As we mentioned before, in our simulation we used GD to minimize the left hand side. After convergence, we computed the top eigenvector $\tilde{q}$ of the Hessian matrix for the obtained $\beta$. Using (S88) below, we calculated the value of the lower bound for $\tilde{q}$. If the objective equaled the lower bound, then we knew we achieved the minimum value of the objective (S82). Interestingly, in all our simulations, this was indeed the case.

To derive a closed form expression for the right-hand side, we use notation and some derivation form Appendix VI, to write

$$
q^T \nabla^2 \mathcal{L}\left(\tilde{\theta}(\beta)\right) q
$$

$$
= \frac{1}{n} \sum_{i=1}^{k} \left[ \left( \sum_{j=1}^{n} q_j x_j I_{j,i} \frac{w_i^{(2)}}{\beta_i} \right)^2 + \left( \sum_{j=1}^{n} q_j I_{j,i} \frac{w_i^{(2)}}{\beta_i} \right)^2 + \left( \sum_{j=1}^{n} q_j \beta_i \sigma \left( w_i^{(1)} x_j + b_i^{(1)} \right) \right)^2 \right] + \frac{1}{n} \left( \sum_{j=1}^{n} q_j \right)^2
$$

$$
= \frac{1}{n} \sum_{i=1}^{k} \left[ \left( \frac{w_i^{(2)}}{\beta_i} \right)^2 \left( \left( \sum_{j=1}^{n} q_j x_j I_{j,i} \right)^2 + \left( \sum_{j=1}^{n} q_j I_{j,i} \right)^2 \right) + \beta_i^2 \left( \sum_{j=1}^{n} q_j \sigma \left( w_i^{(1)} x_j + b_i^{(1)} \right) \right)^2 \right]
$$

$$
+ \frac{1}{n} \left( \sum_{j=1}^{n} q_j \right)^2. \tag{S87}
$$

It is easy to calculate that the minimum of this function w.r.t. $\beta$ over $\mathbb{R}_+^k$, is given by

$$
\min_{\beta \in \mathbb{R}_+^k} q^T \nabla^2 \mathcal{L}\left(\tilde{\theta}(\beta)\right) q =
$$

$$
\frac{2}{n} \sum_{i=1}^{k} \left| w_i^{(2)} \right| \sqrt{ \left( \sum_{j=1}^{n} q_j x_j I_{j,i} \right)^2 + \left( \sum_{j=1}^{n} q_j I_{j,i} \right)^2 } \left| \sum_{j=1}^{n} q_j \sigma \left( w_i^{(1)} x_j + b_i^{(1)} \right) \right| + \frac{1}{n} \left( \sum_{j=1}^{n} q_j \right)^2.
$$

$$
\tag{S88}
$$

We note that in general, the set of implementations over which we optimize can be strictly contained within the set of all implementation, namely

$$
\left\{ \left. \tilde{\theta}(\beta) \,\right|\, \beta \in \mathbb{R}_+^k \right\} \subseteq \Omega(f). \tag{S89}
$$

Therefore,

$$
\min_{\beta \in \mathbb{R}_+^k} \lambda_{\max}\left( \nabla^2 \mathcal{L}\left(\tilde{\theta}(\beta)\right) \right) \geq \min_{\theta \in \Omega(f)} \lambda_{\max}\left( \nabla^2 \mathcal{L}\left(\theta\right) \right). \tag{S90}
$$

In other words, our approach generally provides an upper bound on the sharpness of the flattest implementation. The true value may sometimes be smaller.