# OpenReview forum: "The Implicit Bias of Minima Stability: A View from Function Space"
_NeurIPS.cc/2021/Conference — NeurIPS 2021 Poster_

### Official Review · Reviewer_m6GT · 2021-07-14

**Rating:** 6
**Confidence:** 5

**Summary:**

This paper studies the smoothness of the functions represented by single variate two-layer neural networks at minima with different flatness, and connect this smoothness with the implicit bias of (stochastic) gradient descent to flat minima. Firstly, for a twice differentiable minimum found by SGD, an upper bound for the leading eigenvalue of the loss function's Hessian matrix (with respect to parameters) is derived depending on the learning rate. Then, a weighted L1 norm of the second derivative of the model function (with respect to input data) is upper bounded by the Hessian's leading eigenvalue. Combining these two bounds, the weighted L1 norm of the model function's second derivative is bounded by a quantity depending on the learning rate. This result implies that SGD with bigger learning rate finds flatter minima, and thus finds smoother functions. The result only considers the minima found by optimization algorithms, hence it is independent with initialization.

For neural networks with ReLU activation, minima are sometimes not differentiable. The analysis above is extended to this case and similar bounds are also provided. The theorem works in the mildly over-parameterized case in which the number of neurons is slightly bigger than the number of training data.

Numerical experiments show that GD with bigger learning rate can indeed find smoother functions, especially in the region where training data are densely distributed. This observation is consistent with the theoretical results.

**Limitations And Societal Impact:**

The authors have adequately addressed the limitations. Since this is a theoretical study, I do not think there will be any direct negative societal impact.

**Main Review:**

Flat minima are widely known to have better generalization performance than their sharp counterparts. Many works have explored the connection between algorithms and flatness. Some of them build connection between algorithms and the flatness of the minima they found, while others design new algorithms which can find flatter minima. This work, however, build a more complete theoretical understanding by also addressing another problem: the implicit regularization mechanism of flat minima. The authors show that models at flat minima usually represent smoother functions, which (partially) explains why flat minima generalize better. Exploring the connection between the landscape in the parameter space and the landscape in the input data space seems like a promising way to study the implicit regularization of optimization algorithms.

Some major comments and concerns are:
1. The bounds on \lambda_max only depend on the learning rate, and are independent with batch size, and actually this bound is trivial. Hence, the bounds work for full batch GD and do not distinguish GD and SGD, which means the additional implicit regularization effect of SGD is not shown by the results. In practice, though, SGD is usually observed to have better performance than GD. Bounds for the minima found by SGD depending on both the learning rate and the batch size are derived in existing works, such as [52] cited in the paper. Is it possible to derive similar finer bounds for the smoothness of the model functions?

2. The analysis is conducted in a restrictive setting: single variate two-layer neural networks. This limits the insights provided by and the potential application of the theory. What is the hardness of extending the analysis to high dimensional setting? If theoretical extension is nontrivial, numerical experiments of the high dimensional problem may also show the generality of the theory.

3. On line 135-136, the authors said "in the setting of ReLU networks, this (approximate) common eigenvector assumption holds true" regarding to the assumptions in Lemma 2 above. It there any justification of this claim?

**Time Spent Reviewing:**

3h

---

> ### Author Response · Authors · 2021-08-10
> **Reply to Reviewer m6GT**
>
> Thanks for the constructive comments.
>
> &nbsp;
>
>
> ### **Additional bias of SGD w.r.t. GD:**
>
> Thanks for this great observation. Our theorems characterize the properties of stable solutions. Therefore, we need to apply a necessary condition on stability (in the form of “If $\theta^*$ is a stable minimum, then...”). Unfortunately, the bound in [52] is a sufficient condition and is therefore not suitable for our proof. Currently, we do not know of a necessary condition in the literature that takes both the learning rate and the batch size into account, and deriving such a necessary condition is out of the scope of this paper. This is why our current result does not capture the additional implicit regularization induced by the stochasticity of SGD. Yet, should future work derive a non-trivial necessary condition for SGD stability, based on the top eigenvalue of the Hessian, it will be simple to obtain an improved version of our result (using Lemma 4).
>
> &nbsp;
>
>
> ### **Generalization to higher dimensions:**
>
> This is indeed an interesting extension, which we are currently working on. The analysis of this generalized setting is not trivial, but seems to be at reach. This requires many additional definitions and derivations, and is somewhat similar to the way the one-dimensional results in [R1] were extended to multiple dimensions in [R2]. Particularly, as in [R2], this extension requires the use of the Radon transform. Although this work is in early stages, we are fairly confident that there exists an analogue result for higher dimensions.
>
> As for running higher dimensional simulations, kindly note that we need a generalized version of Lemma 4 (specifically, an integral expression) for higher dimensions, which we currently lack. Therefore, we cannot validate the generalized setting in terms of which solution has better (generalized) smoothness than which.
>
> **References:**
>
> [R1] Savarese et al., “How do infinite width bounded norm networks look in function space?”, COLT\` 19
> [R2] Ongie et al., “A function space view of bounded norm infinite width ReLU nets: The multivariate case”, ICLR\` 20
>
> &nbsp;
>
>
> ### **Where is the justification for the claim of top eigenvector alignment (line 135-136)?**
>
> We are sorry for the missing reference, thanks for catching this. The claim that the top eigenvectors of the Hessians tend to align is based on Lemma 5. We’ll add a pointer to that lemma in lines 135-136. Specifically, it can be seen in Lemma 5 that when the overparameterization is small (i.e. $p_{\max}/n$ is small), the entire right-hand side of Eq. (32) is small, implying that the top eigenvectors of the different systems align. This small overparameterization regime is the one considered later in Theorem 2. We’ll clarify this.

---

### Official Review · Reviewer_d3sA · 2021-07-16

**Rating:** 7
**Confidence:** 4

**Summary:**

This paper studies a single hidden layer univariate ReLU network. The authors show that stable solutions learned by SGD will be functions whose second derivative has bounded weighted L1 norm. Besides, the learned function gets smoother as the learning rate increases.

**Limitations And Societal Impact:**

Yes, the authors adequately addressed the limitations and potential negative social impact of their work.

**Main Review:**

This paper shows that the earned stable solutions will be smoother if we increase the learning rate. Overall this paper is well organized. The results are interesting and seem solid. However, I still have the following concerns:

1.  Because the NN function f is a pice-wise linear function for both input x and parameter \theta. Thus the classical second derivative |f''(x)| doesn't exist. It would be much better if the authors can give the weak definition of |f''(x)| in the main part of the paper rather than in the appendix(line 112). Besides, I get quite confused about the condition |f''(x_j)| < \infty (line 174) because f''(x) is sum of weighted Dirac delta function. The authors may want to add some comments on it.

2. In the definition of linear stability and generalized linear stability(Definition 1 &2), there is a parameter $\epsilon$. Why doesn't $\epsilon$ show up in the condition of Theorem 1&2?

3. Currently, the result of theorem 2 could not hold for the overparameterized setting where the neuron number k > > n, the input dimension also needs to be set as 1.  Could the result of this paper be generalized to the setting without those limitations?

**Time Spent Reviewing:**

5

---

> ### Author Response · Authors · 2021-08-10
> **Reply to Reviewer d3sA**
>
> Thanks for the constructive comments.
>
> &nbsp;
>
> ### **Definition of weak-form derivative of $f(x)$:**
>
> Good point, we will add the weak definition of the second derivative of f to the main text, thanks.
>
> &nbsp;
>
> ### **The condition $|f''(x_j)| < \infty$:**
>
> As written in line 170, the condition $|f''(x_j)| < \infty$ means that the knots of $f$ do not coincide with any training point in the dataset. Namely, this corresponds to assuming that the minimum is twice differentiable. We will clarify this. Note that this requirement is only needed in Theorem 1. It is removed in Theorem 2, which treats minima that are not twice differentiable (where at least one knot of $f(x)$ coincides with some training point). We acknowledge that writing the differentiability condition as $|f''(x_j)| < \infty$ might be confusing, and will therefore write it more explicitly.
>
> &nbsp;
>
> ### **Why doesn’t $\varepsilon$ show up in the conditions of Theorems 1 & 2?**
>
> This is an excellent question, we may not have clarified well enough. $\varepsilon$ has an effect only when the minimum is not differentiable, i.e. when the gradient does not vanish as we approach the minimum. Since in our setting all global minima are differentiable (see Appendix IX.1), and  $\hat{\boldsymbol{g}}^{(t)}_{m} = \boldsymbol{0}$ for all $t$ (see Appendix IX.2 Eq. (S82)), we have that $\gamma$ in Lemma 2 equals zero and thus $\varepsilon$ has no effect. This is not to be confused with the fact that not all global minima are **twice** differentiable.
>
> Perhaps this is best understood through a simple example. Recall we are interested in functions that admit second-order approximations in regions around a (perhaps non-twice-differentiable) minimum. The second-order approximation is all that matters for linear stability, therefore let’s take for simplicity an example where $\mathcal{L}(\theta)$ is either linear or quadratic in each region. Particularly, let us examine two such functions, one differentiable and one not.
>
> First, let’s consider a differentiable function: $\mathcal{L}(\theta)=0.5 \theta^2$ for $\theta<0$ and $\mathcal{L}(\theta)=\theta^2$ for $\theta\geq 0$. In this case, the update rule of gradient descent is $\theta_{t+1}=(1-\eta) \theta_t$ when $\theta_t<0$ and $\theta_{t+1}=(1 - 2 \eta) \theta_t$ when $\theta_t\geq 0$. Note that because the function is differentiable at the minimum $\theta=0$ (i.e. the one-sided derivative at $\theta=0$ is zero at both sides), we have that $\theta_{t+1}$ is proportional to $\theta_t$. Therefore $\theta_t$ can be expressed as $\theta_0 \prod^t_{i = 1} a_i$, where each $a_i$ is either $1-2\eta$ or $1-\eta$. Now, the stability condition says that for all $|\theta_0|<\varepsilon$, we should have $|\theta_t|<\varepsilon$ for all $t$ greater than some $T$. But this translates to the condition that $\prod^t_{i=1} |a_i| < 1$ for all $t>T$, in which $\varepsilon$ doesn’t show up. This example generalizes to the multi-dimensional case, where $\theta_{t+1}$ is a matrix times $\theta_t$ and therefore unrolling of the GD iterates leads to an expression for $\theta_t$ as a product of matrices (rather scalars) times $\theta_0$. In that case as well, $\varepsilon$ has no effect.
>
> Now, let’s consider a non-differentiable function: $\mathcal{L}(\theta)=|\theta|$. In this case the update rule of GD is $\theta_{t+1}=\theta_t- \eta \ \text{sgn}(\theta_t)$. Namely, $\theta_{t+1}$ is not proportional to $\theta_t$. Here, if $\varepsilon<0.5 \eta$ then for any $ \theta_t \in [-\varepsilon,\varepsilon]$, we have that $\theta_{t+1}\notin[\varepsilon,\varepsilon]$  and therefore the minimum $\theta=0$ is not stable for such an $\varepsilon$. On the other hand, if $\varepsilon>\eta$, then it’s easy to see that $|\theta_t|<\varepsilon$ for all $t$. Therefore the minimum $\theta=0$ is stable for such an $\varepsilon$. This exemplifies that in the nondifferentiable case, the stability condition may depend on $\varepsilon$. But, again, this is irrelevant to our setting, in which all global minima are differentiable (even if not twice-differentiable).
>
>
> &nbsp;
>
> ### **Generalization to higher dimensions:**
>
> This is indeed an interesting extension, which we are currently working on. The analysis of this generalized setting is not trivial, but seems to be at reach. This requires many additional definitions and derivations, and is somewhat similar to the way the one-dimensional results in [R1] were extended to multiple dimensions in [R2]. Particularly, as in [R2], this extension requires the use of the Radon transform. Although this work is in early stages, we are fairly confident that there exists an analogue result for higher dimensions.
>
> **References**:
>
> [R1] Savarese et al., “How do infinite width bounded norm networks look in function space?”, COLT\` 19
> [R2] Ongie et al., “A function space view of bounded norm infinite width ReLU nets: The multivariate case”, ICLR\` 20
>
> &nbsp;
>
>
> ### **Generalization to higher over parameterization:**
>
> First, note that the assumption of small over-parameterization is not required in the twice-differentiable case (Theorem 1). It is only required in the more involved non-twice-differentiable setting (Theorem 2). Unfortunately, our current proof technique does not allow us to avoid such an assumption. However, we saw that SGD converges to smooth solutions also when k>>n, as demonstrated in Figure 1. Therefore, we believe that this is an artifact of the proof technique we use, rather than a fundamental limitation. We will add more simulations in this regime for illustration, and will also comment about it in the context of Theorem 2, thanks.

---

> > ### Comment · Reviewer_d3sA · 2021-08-29
> > **Reply to the rebuttal**
> >
> > Thanks for the detailed rebuttal.

---

### Official Review · Reviewer_hfKq · 2021-07-18

**Rating:** 6
**Confidence:** 1

**Summary:**

The goal of this paper is to extend existing results on stable minima found by SGD to the non-differentiable setting of ReLU networks.

In particular, in its main contribution (Theorem 1) the paper shows that the solution found by SGD for 1-dimensional ReLU networks has bounded second derivative, under a weighted total variation norm. Moreover, this weighted norm is controlled by the step-size of SGD.

This implies that also for ReLU networks, SGD with large learning rate is biased towards smooth solutions.

**Limitations And Societal Impact:**

In this paper, a rather simplified setting is considered (1-hidden layer ReLU networks with one-dimensional inputs and outputs). Perhaps it can be mentioned to what extend the results can be generalized to the higher-dimensional setting.

To me, it seems that the analysis does not seem to easily extend to this higher-dimensional setting. In the one-dimensional case, the second derivative of the function can be represented by a sum of Dirac measures, while in the higher-dimensional setting quite complicated scenarios can appear.


**Main Review:**

Strengths:
- The paper is to best of my knowledge the first work which analyzes minima stability in the non-differentiable setting of 1D ReLU networks. I found Theorem 1 a rather interesting and (to me) non-trivial result.
- The theoretical results are to some extend confirmed by the experiments in Fig. 2.

Weaknesses:
- The mathematical analysis framework used to deal with the non-differentiable setting appears not fully rigorous.
- As mentioned in the paper, for ReLU nets f'' is a sum of Dirac measures, so it does not makes sense to talk about f''(x). This is due to the fact that the singular measure f'' does not have a density with respect to the Lebesgue measure dx. The mathematically rigorous formulation of the integral would be \int g(x) d|f''|(x), where |.| denotes the total variation of the measure f''.  In that context, I did not fully understand the meaning of |f''(x_j)| < \infty in Theorem 1. Does it mean that the input data points do not lie at the "jump points" of the function? This would seem like a quite strong restriction.
- A more extensive empirical evaluation/demonstration of Theorem 1 would in my opinion greatly improve the work, e.g., plotting \int g(x) d|f''|(x) for different step-sizes \eta, and comparing it to 1 / \eta.

Comments:
- I believe it needs to be assumed somewhere in Sec. 2.2 that \theta* is in the interior of one of the S_m? Otherwise, its gradient in (11) is not uniquely defined. Most likely this is not a strong assumption, since a tiny perturbation of \theta* should not influence the result.


**Time Spent Reviewing:**

5 hours

---

> ### Author Response · Authors · 2021-08-10
> **Reply to Reviewer hfKq**
>
> Thanks for the constructive comments.
>
> &nbsp;
>
> ### **The condition $|f''(x_j)| < \infty$:**
>
> We would like to clarify that Sec. 2.1 and Theorem 1 are dedicated to the twice differentiable case, in which the knots of $f(x)$ do not coincide with any training point in the dataset. This is concisely captured by the condition $|f''(x_j)| < \infty$, as we explain in line 170. Section 2.2 and Theorem 2 remove this assumption, and treat minima that are not twice differentiable, where at least one knot of f(x) coincides with some training point. We acknowledge that writing the differentiability condition as $|f''(x_j)| < \infty$ might be confusing, and will therefore write it more explicitly.
>
> &nbsp;
>
> ### **Integral notation:**
>
> We agree that the correct formulation of the integral is via the total variation of the measure $f''$. Our simplified notation follows prior works in this area (e.g., [R1]). However, to be more rigorous, we will add a comment on how one should interpret this notation, thanks.
>
> &nbsp;
>
> ### **More extensive empirical demonstration of Theorem 1:**
>
> This is a good point. We’ll add simulations with varying step-sizes $\eta$, and different initializations, and will plot the resulting integral of Theorem 1. We believe this will nicely support our theoretical observations. Thanks.
>
> &nbsp;
>
> ### **Should $ \theta^{*} $ be assumed to lie in the interior of one of the $\mathcal{S}_m$? Otherwise, its gradient in (11) is not uniquely defined.**
>
>
> Please note that Sec. 2.2 is dedicated to non-differentiable minima. Therefore, the gradient of $\mathcal{L}$ at $\theta^*$ indeed may not be uniquely defined. The purpose of Sec. 2.2 is precisely to investigate this setting. Had $\theta^*$ lied in the interior of one of the $\mathcal{S}_m$, it would be a twice differentiable minimum, and there would not be a reason to discuss these regions in the first place, since this is exactly the setting we discuss in Sec 2.1.
>
> &nbsp;
>
> ### **Generalization to higher dimensions:**
>
> This is an interesting point, which we are currently working on as an extension. The analysis of this generalized setting is indeed not trivial, but seems to be at reach. This requires many additional definitions and derivations, and is somewhat similar to the way the one-dimensional results in [R1] were extended to multiple dimensions in [R2]. Particularly, as in [R2], this extension requires the use of the Radon transform. Although this work is in early stages, we are fairly confident that there exists an analogue result for higher dimensions.
>
> **References**:
>
> [R1] Savarese et al., “How do infinite width bounded norm networks look in function space?”, COLT\` 19
> [R2] Ongie et al., “A function space view of bounded norm infinite width ReLU nets: The multivariate case”, ICLR\` 20

---

### Official Review · Reviewer_g3RQ · 2021-07-22

**Rating:** 7
**Confidence:** 3

**Summary:**


This paper attempts to answer the implicit bias of SGD via a stability analysis on a single hidden layer univariate ReLU network. The main result is that a stable solution that can be found by a larger learning rate must be smoother. In detail, they show the weighted L1 norm of the second derivative is upper bounded by roughly $2/\eta$, where $\eta$ is the learning rate.

**Limitations And Societal Impact:**

Yes

**Main Review:**

This paper presents an interesting result about implicit regularization of SGD, that the stable local minimum which can be found by SGD must be smooth enough, depending inversely on the learning rate. The key observation is that for this specific single layer univariate network, the spectrum of the hessian at interpolating solution forms an upper bound for the weighted integral of the absolute value of the second derivative with respect to the input. One main technical contribution is that the authors also manage to deal with the case of non-smooth case. Experiments in Figure 1 also suggest the theory in this paper captures the phenomena in practical training pretty well.

However, the analysis in the paper doesn't give a complete story for the implicit bias of SGD, because a priori, it's not clear if SGD is always converging to a linearly stable solution. Moreover, it's not even clear if SGD is converging to any solution. On the other hand, note both the gradient and its noise vanish at any interpolation solution, thus even if a solution is not stable, SGD will not necessarily deviate it without stronger noise assumptions, which is non-vanishing at global minimum. In the analysis of the paper, the authors actually don't use any property about the noise, no matter if it's SGD noise, isotropic noise, or even just noiseless full-batch GD. The stability result still applies.

Therefore, the main message of this paper should be interpreted as, strong stability solution of SGD with a large learning rate is smooth but not "the function implemented by the network upon convergence gets smoother as the learning rate increases", as stated in the abstract, which is kind of overselling.

I also have a question about the claim that "a trained model tends to be smoother at the center of the training distribution". Is this verified by experiments or only from the upper bound? As far as I can see, the derivation of the upper bound is quite crude (e.g. fixing $u$ to be all one vector) and use $\min(g^+,g^-)$ to lower bound $g^+$ or $g^-$ and there's room to further tighten the bound. For example, if the authors don't use $\min(g^+,g^-)$ but $g^+$ or $g^-$ directly, does the weight in the bound still peak in the middle?

Overall, I found the idea of this paper quite novel and interesting. The writing is also clear and the paper is easy to follow. Thus I tend to accept the paper and would like to improve the score if the authors could give satisfactory answers to my above questions.


**Time Spent Reviewing:**

5

---

> ### Author Response · Authors · 2021-08-10
> **Reply to Reviewer g3RQ**
>
> Thanks for the constructive comments.
>
> &nbsp;
>
> ### **SGD doesn’t necessarily converge to a linearly stable minimum:**
>
> Thanks for this comment. It is very important and we will try to better clarify it in the paper.
>
> It is true that if SGD is initialized precisely at a global minimum (in our overparameterized setting), then it stays there whether it is stable or not. However, with random initialization, this event (“initialization at a global minimum”) happens with probability 0. As for arriving precisely at a global minimum at some later time during training, or converging to it, this also seems to happen with probability zero. Specifically, we do not have a formal proof for this, but to the best of our knowledge, SGD was never observed in a practical NN setting to converge to a linearly unstable minimum. This can be seen, for example, in [R1], which carried out extensive experiments on SGD training. Their results show that the top eigenvalue of the Hessian of the minimum to which SGD converges, is always $\leq 2/\eta$ (see also [R2] for additional linear stability results and experiments). Therefore, in practice, SGD does converge only to linearly stable solutions.
>
> **References**:
>
> [R1] Cohen et al., “Gradient Descent on Neural Networks Typically Occurs at the Edge of Stability”, ICLR\`  21
> [R2] Wu et al., “How SGD Selects the Global Minima in Over-parameterized Learning: A Dynamical Stability Perspective”, NeurIPS\` 18
>
> &nbsp;
>
> ### **SGD doesn’t necessarily converge to a global minimum:**
>
> This is certainly true, but kindly note that this does not contradict our results. We only characterize the global minima to which SGD **can** converge. Namely, all we state are results of the form “if SGD has converged to a global minimum, then...”. Note that in practice, it is fairly simple to verify that a solution is not a global minimum (does not interpolate the training points), in which case practitioners often do not use the trained model and rather try different training parameters. Therefore, characterizing global minima is of practical importance, as opposed to local minima and saddle points, which are typically not used in practice.
>
> &nbsp;
>
> ### **A trained model tends to be smoother at the center of the training distribution:**
>
> We indeed verified this behavior in simulations, as demonstrated e.g. in Figure 1. We’ll add more simulations to the supplementary to further illustrate this.
>
> &nbsp;
>
>
> ### **Crudeness of upper bound:**
>
> Thanks for your careful reading of our proofs!
>
> Regarding choosing $u$ to be an all-ones vector, note that the matrix $\Phi$  (for which we bound the top singular value) has the neural tangent features at its columns. It was shown that during training, the neural tangent features tend to align [R3], which implies that their top singular vector is close to an all-ones vector. Therefore, setting $u$ to be proportional to an all-ones vector is usually quite reasonable (yet not necessarily optimal). We agree that there might be some room for improving the tightness of this choice.
>
> Regarding $g^+$ and $g^-$, this is a great question. Note that any function $f$ implemented by a network, can be implemented in infinitely many ways (i.e. with different sets of parameters). When $f$ corresponds to a global minimum, some of these implementations may correspond to flatter global minima, and some to sharper global minimima. Recall that our goal is to find a lower bound on the top eigenvalue of the Hessian for **any implementation** of $f$ (see lines 219-225 for motivation and explanation). Now, choosing between $g^+$ and $g^−$ directly according to the sign of $w^{(1)}$, is implementation dependent, and therefore not good for our purposes. Namely, there may be alternative implementations of the same function $f$, in which $w^{(1)}$ has different sign patterns. And a bound that applies to only one sign pattern does not necessarily apply to a different sign pattern. Our choice of using $\min(g^+,g^-)$ guarantees that the bound applies to **all** possible implementations of the function $f$.
>
> Note that had we chosen directly $g^+$ or $g^-$ based on the sign of $w^{(1)}$, each possible implementation of $f$ would correspond to a different weighting function, with peak at a possibly different location. When we use our bound on the flattest implementation, $\min(g^+,g^-)$, the weighting function has its peak approximately at the middle, as we illustrate in Fig. 2.
>
> It should be noted that when $f$ has fewer knots than the number of neurons in the network (i.e. slight redundancy of neurons), there exists an implementation of $f$ for which $\frac{1}{n^2} || \Phi \boldsymbol{1} ||^2$ equals our lower bound. In other words, in certain cases our approach of taking $\min(g^+,g^-)$ is tight.
>
> **References**:
>
> [R3] Baratin et al., “Implicit Regularization via Neural Feature Alignment”, AISTATS\` 21

---

> > ### Comment · Reviewer_g3RQ · 2021-08-31
> > **Reply to author response**
> >
> > Thanks for the response. The explanation for why can't replacing $\min(g^+,g^-)$ by $g^+$ or $g^-$ makes sense and I increased my score by one point. I thought there's an easy way to use the sign of f''(x) but not only |f''(x)| to get a tighter bound via a better choice of $g^+$ or $g^-$, but a naive analysis won't distinguish $relu(x)$ and $-relu(-x)$ and the boun indeed seems to be the best one can get with the current proof.

---

### Decision · Program_Chairs · 2021-09-27

**Decision:**

Accept (Poster)

**Comment:**

This work analyzes the prediction surface of weights reached by descent methods, arguing it is smooth on average (integral of second derivative is small).  Reviewers are supportive, but had many detailed comments, to which the authors in turn gave detailed responses.
 I believe incorporating these discussions and comments in revisions will greatly strengthen the paper, and urge the authors to do so.